# Gain engineering and atom lasing in a topological edge state in synthetic dimensions

Takuto Tsuno[1,4], Shintaro Taie [1,4], Yosuke Takasu [1] ✉, Kazuya Yamashita[1,3], Tomoki Ozawa [2] & Yoshiro Takahashi [1]

Recent advances in quantum technology have highlighted the importance of controlling quantum states, especially in open quantum systems, where the system interacts with the environment. Non-Hermitian quantum mechanics describes these systems. Photonic systems are a key platform for studying non-Hermitian quantum mechanics owing to their ability to engineer gain and loss. Ultracold atomic gases also have been used to study non-Hermitian quantum mechanics; however, unlike photonics, gain control is challenging, limiting exploration to control of loss. In this paper, we report engineering of effective gain through evaporative cooling of judiciously selected initial thermal atoms, leading to Bose–Einstein condensation (BEC) in the excited eigenstates of a synthetic lattice. We achieve BEC formation in a topological edge state of the Su–Schrieffer–Heeger lattice in the synthetic hyperfine lattice, akin to atomic laser oscillations at a topological edge mode, that is, a topological atom laser.

Controlling quantum states has become crucial in quantum technology, particularly for open quantum systems coupled to the environment[1]. One important framework in describing open quantum systems is the non-Hermitian quantum mechanics[2,3]. Dating back to the study of nuclear decay[4], non-Hermitian quantum mechanics has been rediscovered and studied in various contexts. One turning point is the discovery of parity-time reversal($\mathcal{PT}$)-symmetric Hamiltonians[5,6], which, despite being non-Hermitian, give rise to real energy spectrum. Another very recent development came from understanding of the topological properties of non-Hermitian Hamiltonians[7,8]. Since non-Hermitian Hamiltonians generally have complex eigenenergies, modes with positive imaginary part can lead to amplification and lasing. Topological laser, which is a laser whose lasing mode is a topological edge mode, has been realized[9–15], and its potential application in integrated photonics has been actively pursued[16]. In quantum many-body systems coupled to the environment, an effective description in terms of non-Hermitian Hamiltonians is possible in the quantum trajectory approach between quantum jumps[17,18], and the effect of non-

Hermiticity in many-body quantum phases have been actively studied[19–33].

Since its experimental realization of $\mathcal{PT}$-symmetric optics in 2009[34–36], non-Hermitian quantum mechanics has been actively studied in optics[37–39]. Despite the huge success, optical systems have a limitation that the interaction is typically classical nonlinearity, making it difficult to study quantum many-body physics in non-Hermitian setups. An emerging platform to study non-Hermitian quantum mechanics is ultracold atomic gases, where one can study quantum many-body physics in a controlled manner[40–42]. Recently, there have been experiments in which non-Hermitian Hamiltonians are realized by controlling dissipation in ultracold atomic gases[43,44]. However, ultracold atomic gases have their own limitation that the non-Hermiticity that can be introduced has been limited to the control of dissipation. It has thus not been possible to study non-Hermitian physics related to controlled gain in ultracold atomic gases. Without gain, it is not possible to study phenomena related to amplification and lasing.

[1]Department of Physics, Graduate School of Science, Kyoto University, Kyoto 606-8502, Japan. [2]Advanced Institute for Materials Research (WPI-AIMR), Tohoku University, Sendai, Miyagi 980-8577, Japan. [3]Present address: Center for Quantum Information and Quantum Biology, The University of Osaka, Toyonaka, Osaka 560-0043, Japan. [4]These authors contributed equally: Takuto Tsuno, Shintaro Taie. ✉e-mail: takasu@scphys.kyoto-u.ac.jp

Here, we report engineering of gain in ultracold atomic gases and observe the formation of Bose–Einstein condensation (BEC) in an excited eigenstate of a synthetic hyperfine lattice. The gain is provided by evaporative cooling of appropriately chosen initial thermal atoms. We couple ground-state hyperfine levels of $^{87}$Rb atoms by microwaves to realize a lattice along a synthetic dimension made of hyperfine states[45–49]. We show that a BEC can be formed in a dark state of a three-site lattice by preparing thermal atoms, using STimulated Raman Adiabatic Passage (STIRAP)[50,51], in the coherent superposition of two sites which form the dark state. Note that the concept of a dark state is crucially important for novel phenomena in quantum optics like electromagnetically induced transparency[52] and lasing without inversion[53,54], and also plays an essential role in realizing a localized state in a flat-band in condensed matter systems[55] and spatial adiabatic passage in matter-wave optics[56]. We furthermore realize a five-site Su–Schrieffer–Heeger (SSH) chain[57] and realize the formation of a BEC in its topological edge state. Formation of a BEC in an atomic gas has a close analogy to laser oscillation and sometimes referred to as atom laser[58]. We have thus demonstrated lasing of atomic matter wave in a topological edge state, namely a topological atom laser. Our results make ultracold atomic gases a promising platform to study non-Hermitian quantum mechanics with both gain and loss, opening an avenue toward controlled study of topological atom optics and quantum many-body physics in non-Hermitian setups.

## Results
### Experimental results
We achieve gain engineering in a lattice made of a synthetic dimension using ground-state hyperfine sublevels of $^{87}$Rb. Here, we utilize long-lived nature of the $^{87}$Rb hyperfine states in the electronic ground state to observe the formation of a BEC in the synthetic lattice. The scattering length of $^{87}$Rb exhibits only a small dependence on hyperfine or magnetic sublevels[59,60]. As a result, the spin-exchange collisions are significantly suppressed. By coupling hyperfine sublevels $|F (= 1, 2), m_F\rangle$ in the ground $5s\ ^2S_{1/2}$ state of $^{87}$Rb with microwaves, we prepare 3-site and 5-site lattices in a synthetic dimension, as shown in Fig. 1a,b.

Our experiments start with $^{87}$Rb atoms above the critical temperature $T_c$ for BEC trapped in the crossed optical dipole trap at about 1064 nm[61]. The subsequent microwave sequence consists of two stages: First, we apply time-dependent microwaves to prepare thermal atoms in a superposition of the synthetic lattice sites close to one edge. Then, further evaporative cooling is performed with microwave couplings kept constant (see Fig. 1c, d). Resulting formation of a BEC corresponds to lasing of atoms at a specific eigenmode in the lattice. We properly choose initial thermal atoms so that the resulting BEC is formed not at the lowest-energy mode or a single hyperfine state but rather at an excited mode made of a coherent superposition of multiple hyperfine states. The resulting site distributions are measured by applying Stern–Gerlach separation immediately after switching off the trap, as shown in Fig. 1e.

We first perform an experiment with a 3-site lattice to demonstrate our ability to control the formation of BEC into a dark state, which is not the ground state of the 3-site lattice. Here, we exploit $\Lambda$-type coupling scheme with temporal evolution known as STIRAP. Three levels $|1, 1\rangle$, $|2, 0\rangle$ and $|1, -1\rangle$ are coupled by microwaves and construct a 3-site synthetic lattice (site 4, 3 and 2). The site indices are labeled to be consistent with the 5-site lattice we introduce later. The Hamiltonian in the rotating frame, in the matrix form, is

$$\mathcal{H} = \frac{\hbar}{2}\begin{pmatrix} 2\delta & \Omega_1 & 0 \\ \Omega_1 & 0 & \Omega_2 \\ 0 & \Omega_2 & -2\delta \end{pmatrix} \tag{1}$$

where $\Omega_1, \Omega_2$ are the Rabi frequencies of microwave couplings, and $\delta$ is the detuning due to a magnetic field fluctuation. If the two-photon resonance $\delta = 0$ holds, a dark state, given by $\cos\theta|1, 1\rangle + \sin\theta|1, -1\rangle$ with $\tan\theta = \Omega_1/\Omega_2$, becomes one of the eigenstates of Hamiltonian (1). By applying two microwaves in counter-intuitive order so that $\theta$ changes from 0 to $\pi/2$, the dark state can adiabatically evolve from $|1, 1\rangle$ (site 4) into $|1, -1\rangle$ (site 2). This process is well-known as STIRAP. We utilize this STIRAP process to prepare thermal atoms in a coherent

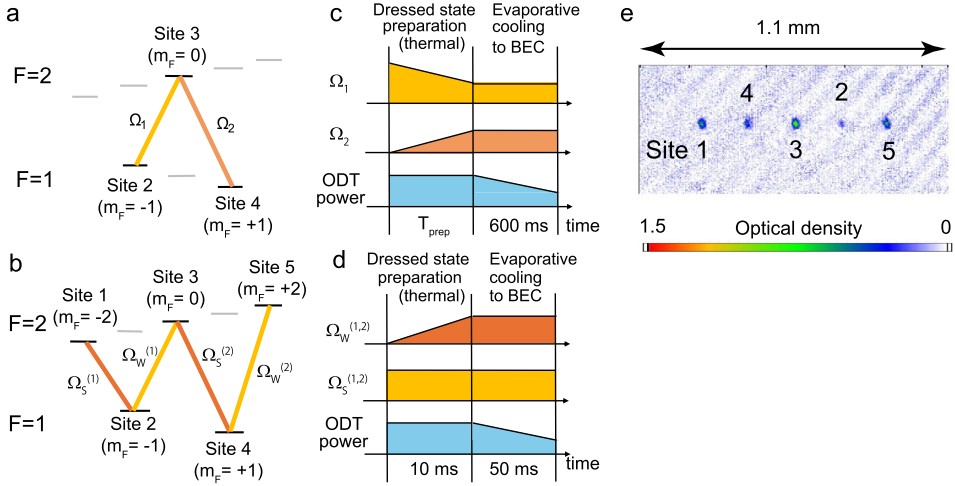

**Fig. 1 | Schematic of the experiment.** Microwave coupling scheme. The 3-site lattice with $\Lambda$-type coupling (**a**) and the 5-site lattice with $W$-type coupling (**b**) are shown. Rabi frequencies for the microwave coupling are indicated by $\Omega_i(i = 1, 2)$ for 3-site lattice and $\Omega_j^{(i)}(i = 1, 2, j = W, S)$ for 5-site lattice. Each state is named as site 1-5 of the synthetic lattice. **c**, **d** Experimental sequence. For both 3- and 5-site experiments, thermal atoms are loaded with microwaves of varying Rabi frequencies to create a superposition and then cooled to achieve BEC. The 3-site sequence **c** is parameterized by the final values of Rabi coupling (not shown in the figure) where $T_{prep}$ is the time required for STIRAP. At the time $T_{prep}$ after the

microwave sequence starts, Rabi frequencies are kept constant and atoms with superposition are cooled to BEC. Similarly, the 5-site sequence **d** is characterized by the final ratio of $\langle\Omega_W\rangle/\langle\Omega_S\rangle$. $\langle\Omega_i\rangle(i = W, S)$ is defined as averaged Rabi frequencies of the same kind of microwaves. ODT means optical dipole trap. **e** Site-resolved imaging by Stern–Gerlach separation. The image is taken after 8.5 ms of free expansion. The order of sites 1, 4, 3, 2, 5 in the absorption image is not a simple ascending order because of the difference in the sign of the magnetic moments of the states between the $F = 1$ and $F = 2$ states.

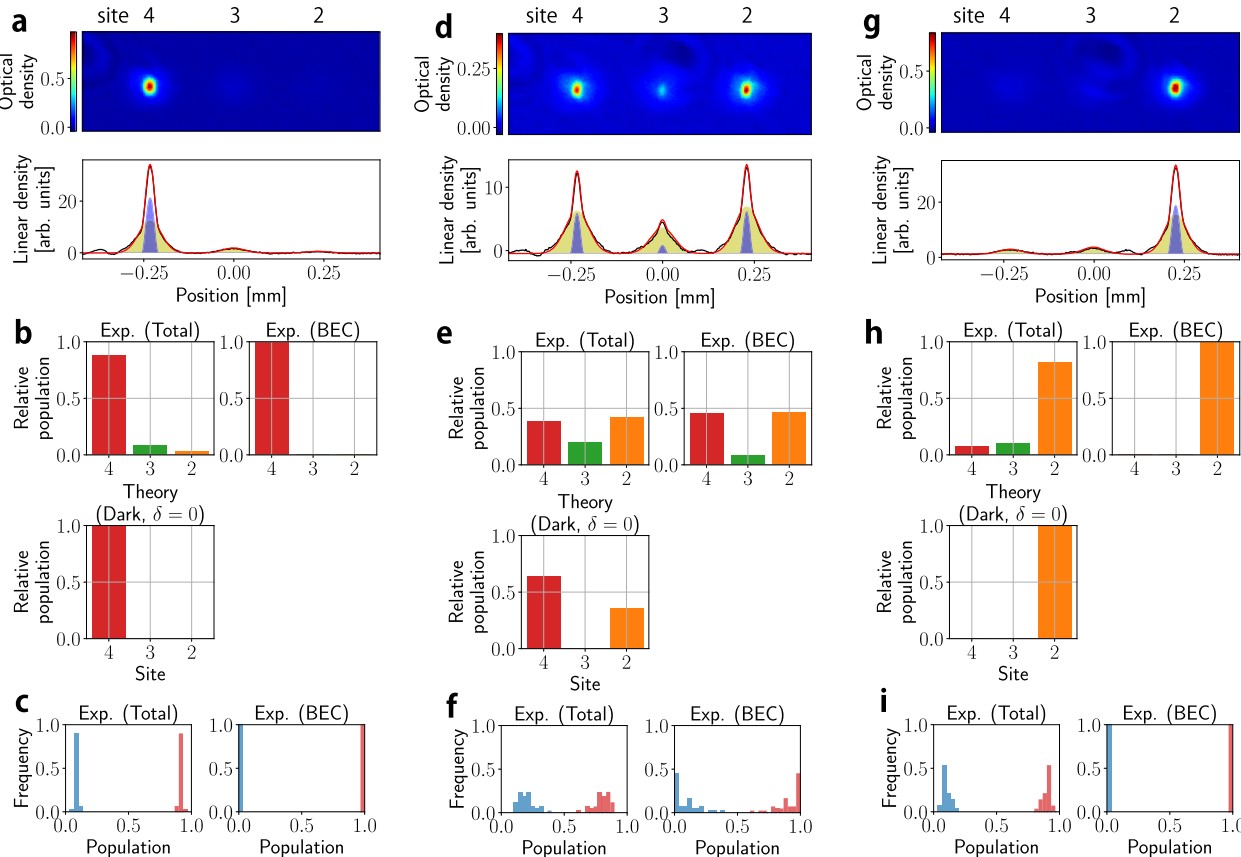

**Fig. 2 | Gain engineering in the 3-site synthetic lattice.** Site distributions after the 3-site sequence with **a**–**c** $\Omega_1/\Omega_2 = 19.27$, **d**–**f** $\Omega_1/\Omega_2 = 1.345$, **g**–**i** $\Omega_1/\Omega_2 = 0.02870$. The figures in **a**, **d**, **g** are typical site-resolved images (time of flight is 8.5 ms) and the corresponding column densities (black lines) with the results of bimodal fitting (red lines), where thermal and BEC components are indicated by the yellow and blue shaded area, respectively. BEC fractions are **a** 0.306(2), **d** 0.156(1), **g** 0.223(1), respectively. The histogram in **b**, **e**, **h** shows the relative population corresponding to the images in (**a**, **d**, **g**). The panel titled Theory (Dark, $\delta = 0$) shows a site superposition of synthetic lattice sites. Initially, atoms are evaporatively cooled in the state $|1,1\rangle$ (site 4). At $T/T_c \sim 1.1$, evaporation is paused, and the STIRAP sequence is applied. The STIRAP with full sweep ($\theta = 0 \rightarrow \pi/2$) takes 10 ms. In our experiment, the actual sweep time $T_{\mathrm{prep}}$ is dependent on the final value of $\theta$: e.g., $T_{\mathrm{prep}} = 5$ ms leads to the almost equal superposition of site 4 and 2 ($\tan \theta = \Omega_1/\Omega_2 = 1.345$). After the sweep, we keep the microwave coupling and restart evaporation to form BEC, followed by the site occupation measurement. As we will show, the BEC is predominantly formed in the dark state where the thermal atoms are prepared.

Figure 2 shows the site occupancies $N_4$, $N_3$ and $N_2$ as well as the number of Bose-condensed atoms $N_{4c}$, $N_{3c}$, and $N_{2c}$, measured at three different $\Omega_1/\Omega_2$ (see Methods). Figure 2a–c shows the case of $\Omega_1/\Omega_2 = 19.27$ where atoms almost localize in the site 4. By choosing different values of $\Omega_1/\Omega_2$, the atoms can be transferred to the site 2 (Fig. 2g–i) or in between the two sites as in Fig. 2d–f. The success of the STIRAP between the sites 4 and 2 indicates the coherent superposition is maintained during the process. In Fig. 2b, e, h, we plot the number of atoms (Bose-condensed atoms) in sites 2 and 4 versus site 3 normalized by the total number of atoms (Bose-condensed atoms) for each performed experiment. The absence of particles in the intermediate state $N_3$ is the indication that the atoms are in the dark state. Small but nonzero $N_3$ is caused by the deviation of the microwave frequencies from the resonances.

distribution of the dark state calculated from diagonalizing the Hamiltonian in the Eq. (1) with $\delta = 0$. The graphs in **c**, **f**, **i** are statistical distributions taken over at least 30 experimental runs. The red bars in the Exp.(Total) (Exp.(BEC)) in **c**, **f**, **i** show $(N_2 + N_4)/N$ $((N_{c,2} + N_{c,4})/N_c)$, where $N_i(N_{c,i})$ indicates the total (condensate) atom number in the site $i$ ($i = 2, 3, 4$). $N = \sum_i N_i$ and $N_c = \sum_i N_{c,i}$. The blue bars in the Exp.(Total) (Exp.(BEC)) show $N_3/N$ ($N_{c,3}/N_c$). Source data are provided as a Source Data file.

We note that the dark state where the condensate is formed is not the ground state of the lattice. This formation of BEC at an excited state can be understood by the difference of the critical temperatures; among the three dressed states, the dark state had most thermal atoms and thus has the highest BEC transition temperature, resulting in preferred formation of the BEC.

We can alternatively describe this process in terms of different effective gain for each lattice site. Although evaporative cooling acts on all sites equally, the initial nonequal distribution of thermal atoms results in different effective gain for each site. Sites with more thermal atoms imply larger effective gain, and since the thermal atoms are initially distributed to the sites forming the dark state, the preferred lasing, namely BEC, takes place at the dark state. This perspective on effective gain is more quantitatively discussed later for the case of 5-site SSH model, comparing the experiment with theoretical modeling involving effective gain. We note that formation of BEC in a dark state of momentum states was recently reported[62] through a kind of mode-conversion by coupling atoms already in a BEC to an optical cavity. Our approach, instead, provides a direct means to control gain in ultracold atomic setups.

Next, we realize a 5-sites SSH lattice[57] to obtain the topological edge-state lasing, or the topological atom laser. The SSH model is a one-dimensional lattice model with alternating hopping strengths,

whose Hamiltonian in the rotating frame is

$$H_{\text{SSH}} = \hbar \left( \frac{\Omega_A}{2} \sum_i^N |2i\rangle\langle 2i-1| + \frac{\Omega_B}{2} \sum_i^N |2i+1\rangle\langle 2i| \right) + \text{H.c.,} \quad (2)$$

where the sum of $i$ runs over integer values. The energy bandgap closes at $\Omega_A = \Omega_B$, and the regions $\Omega_A > \Omega_B$ and $\Omega_A < \Omega_B$ are topologically distinct. Due to the bulk-edge correspondence, when a lattice is terminated at $i = 1$ by restricting the above sum to $i \geq 1$, there exists a zero-energy edge state localized around the edge if and only if $\Omega_A < \Omega_B$, and this zero-energy state has nonzero amplitude only on odd sites (1, 3, 5, $\cdots$ ). The existence of the zero-energy edge-localized state is topologically robust; small spatial inhomogeneity in $\Omega_A$ and $\Omega_B$ will not destroy the edge state. As we see in Methods, such inhomogeneity is naturally present in our setup and thus the topological atom laser we realize is directly benefited from the topological robustness.

In our experiment, five levels $|2, -2\rangle, |1, -1\rangle, |2, 0\rangle, |1, 1\rangle$ and $|2, 2\rangle$ are coupled by microwaves and construct a 5-site synthetic lattice (site 1-5, respectively) for which the Hamiltonian in the rotating frame can be written in the matrix form

$$\mathcal{H} = \frac{\hbar}{2} \begin{pmatrix} -4\delta & \Omega_S^{(1)} & 0 & 0 & 0 \\ \Omega_S^{(1)} & 2\delta & \Omega_W^{(1)} & 0 & 0 \\ 0 & \Omega_W^{(1)} & 0 & \Omega_S^{(2)} & 0 \\ 0 & 0 & \Omega_S^{(2)} & -2\delta & \Omega_W^{(2)} \\ 0 & 0 & 0 & \Omega_W^{(2)} & 4\delta \end{pmatrix} \quad (3)$$

where $\Omega_W^{(1,2)}$ and $\Omega_S^{(1,2)}$ are the Rabi frequencies in microwave couplings, and $\delta$ is the detuning due to a magnetic field fluctuation. The ideal SSH model is obtained if $\Omega_i^{(1)} = \Omega_i^{(2)}$ for $i$ = S, W, and the edge state localized around site 5 exists when $\Omega_W < \Omega_S$; such an edge state is a superposition of only three sites, (sites 5, 3 and 1) if $\delta = 0$. In our setup, $\Omega_i^{(1)}$ and $\Omega_i^{(2)}$ are not exactly equal because of the different Clebsch-Gordan coefficients. As we noted above, the existence of a topological edge state is robust against such inhomogeneity or disorder in hopping amplitudes. Below, we use the averaged values $\langle\Omega_i\rangle = (\Omega_i^{(1)} + \Omega_i^{(2)})/2$ for these couplings to characterize the strong and weak couplings. (See Methods for more details).

To achieve BEC in the topological edge state, thermal atoms initially populated in the site 4 are first transferred to the site 5 by adiabatic rapid passage (ARP). Four microwaves $\Omega_W^{(1,2)}$, $\Omega_S^{(1,2)}$ are then applied as shown in Fig. 1b. Thermal atoms will then adiabatically populate the topological edge mode extending over sites 5, 3, and 1. Finally, we keep the microwave coupling and restart evaporation to accomplish BEC, followed by the site occupation measurement. We performed experiments with three values of the final ratios $\langle\Omega_W\rangle/\langle\Omega_S\rangle$ ( < 1) to confirm the behavior of the topological edge-state lasing.

Figure 3a, d, f shows the time-of-flight signals after the Stern–Gerlach separation, revealing site occupancies $N_i$ ($i$ = 1, ..., 5) at three different values of $\langle\Omega_W\rangle/\langle\Omega_S\rangle$. We see that in all three values of $\langle\Omega_W\rangle/\langle\Omega_S\rangle$, the BEC phase transition, namely, atom lasing is successfully observed, and the formed BEC has the largest population in site 5, with certain atom numbers in 3 and 1 with almost no occupation in 2 and 4, which is consistent with the wavefunction profile of the topological edge state of the SSH model. In the Fig. 3c, f, i, we plot the relative atom number of odd sites versus atom number of even sites. The histograms indicate that the population in sites 2 and 4 are small as expected. If the value of the diagonal components in the Hamiltonian shown in Eq. (1) is all zero, chiral symmetry is preserved and therefore, population in sites 2 and 4 should be zero for the topological edge state.

In our system, the stability of the state preparation is limited by the existence of a magnetic field fluctuation and the uncertainty in determining microwave resonances. Both of them can cause a two-

photon detuning, which eliminates the edge state (See Methods for the stabilization of magnetic field).

## Theoretical model

In order to verify that the formation of the BEC in the edge state is due to the effective gain in the system, we construct a theoretical model including gains, losses and magnetic fluctuation for a 5-site model. We simulate the formation of the condensate $|\psi\rangle$ by numerically solving the nonlinear non-Hermitian Schrödinger equation

$$i\hbar \frac{d}{dt} |\psi\rangle = \left( \hat{H}_0 + \hat{H}^{\text{gain}}(n_T) + \hat{H}^{\text{loss}} \right) |\psi\rangle. \quad (4)$$

Here, $\hat{H}_0$ represents the Hermitian Hamiltonian dynamics of Eq. (3) with the magnetic field fluctuation as well as the time dependence of the Rabi couplings. $\hat{H}^{\text{gain}} = i\hbar g n_T |T\rangle\langle T|$ represents the gain term with the coefficient $g$ being the effective strength of the gain per unit thermal component. We denote the (coherent) spin components of the thermal atoms as $|T\rangle$. The strength of the gain is multiplied by the number of atoms in the thermal components, which we denote by $n_T$. Note that the gain is related with the microscopic process of bosonic stimulation through atom collisions and thus, the gain coefficient $g$, introduced as a phenomenological parameter, should be multiplied by the number of thermal atoms, which is how we constructed the theoretical model. $\hat{H}^{\text{loss}}$ represents the state-dependent two-body inelastic loss of $^{87}$Rb atoms. Details of the theoretical modeling is also given in the Methods.

Figure 4 shows the numerically obtained dynamical formation of the BEC, which corresponds to the parameters of Fig. 3d–f ($\langle\Omega_W\rangle/\langle\Omega_S\rangle = 0.23$). Figure 4a, b shows the time evolution of the site distribution of **a** condensed atoms and **b** thermal atoms, respectively. Figure 4c, d shows the relative population so that the sum of the relative population is 1 at each time. Here, the origin of time (0 ms) is when the gain is introduced, and the previous time $t$ (<0 ms) corresponds to the time when the adiabatic preparation of the system takes place. After the site preparation in 10 ms, the thermal populations (Fig. 4d) are nearly consistent with the expected edge states obtained by diagonalization of the Hamiltonian (Fig. 4e), indicating that the edge state are adiabatically prepared. The small oscillation of population indicate the influence of an external oscillating magnetic field of 0.1 mG.

After gain is introduced at $t = 0$ ms, the site distribution of the condensed atomic cluster gradually approaches the edge state. Figure 4f plots $(N_{c,1} + N_{c,3} + N_{c,5})/N_c$ as a parameter indicating how close to the edge state it is. As expected, atoms are present only at odd-numbered sites, consistent with the zero-energy edge mode originated from the chiral symmetry. Figure 4g shows the time dependence of fidelity $|\langle\psi(t)|\Psi\rangle|/\sqrt{|\langle\psi(t)|\psi(t)\rangle\langle\Psi|\Psi\rangle|}$, where $|\psi(t)\rangle$ is obtained from numerical calculation, $|\Psi\rangle$ is the expected edge states shown in **e**. We can see that the fidelity approaches one, which indicates that the final state is the expected edge state. The relative populations and fidelity for the BEC shown in Fig. 4c, f, g are only important when the BEC fraction is increased to a certain amount beyond the initial seed necessary in the numerical calculation. Thus, our experimental observation of edge-state formation of BEC can be reproduced and explained by the theoretical model with an effective gain, taking into account the realistic magnetic field fluctuations and two-body loss.

## Confirming the coherence of the topological atom laser

We have moreover performed an additional experiment to confirm coherence of the superposition of spin states in the created BEC as shown in Fig. 5. After loading atoms into the lattice with largest ratio of $\langle\Omega_W\rangle/\langle\Omega_S\rangle$, we apply the reverse process back to the $\langle\Omega_W\rangle/\langle\Omega_S\rangle = 0$. The resulting site population returns to the initial site 5, proving that the

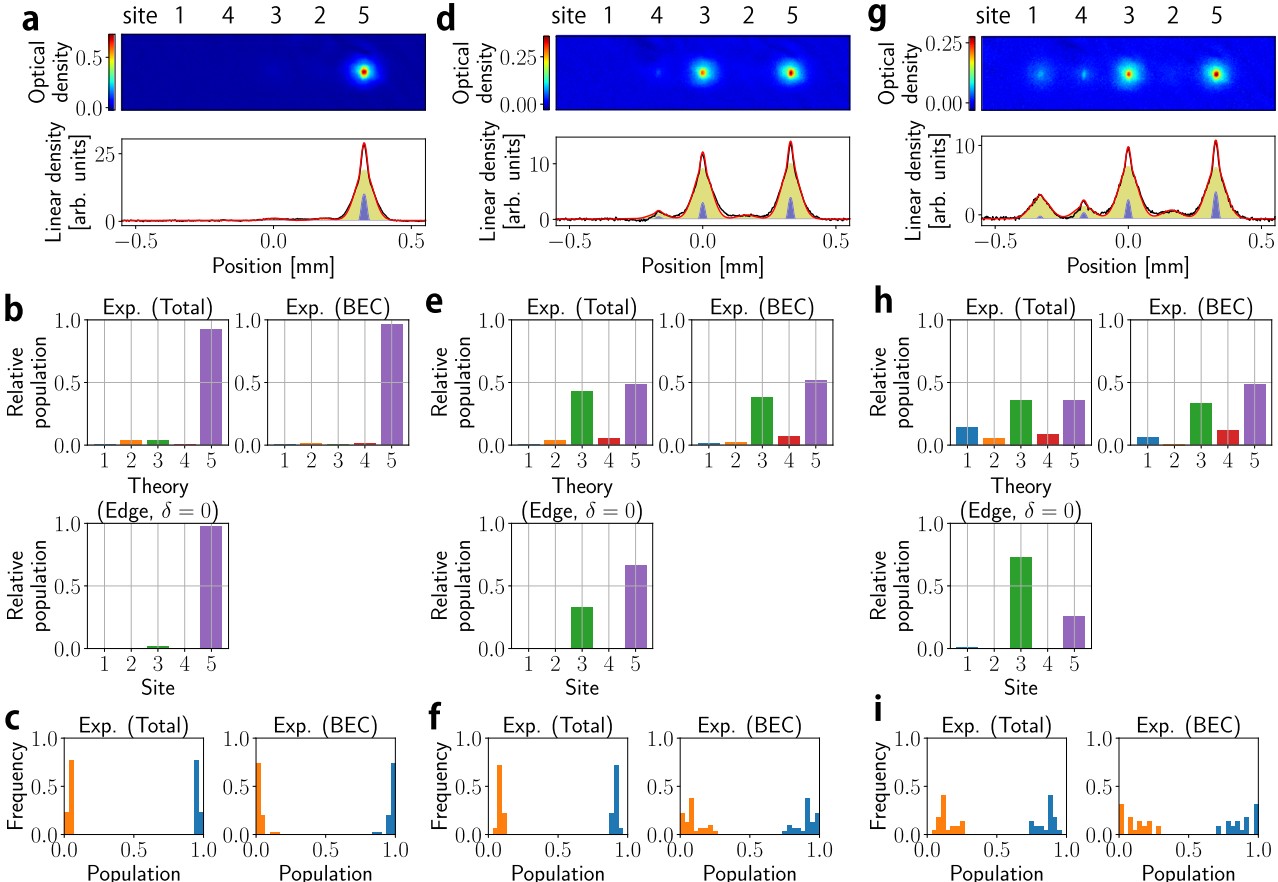

**Fig. 3 | Gain engineering for topological edge states in the 5-site SSH model.** The data shown in (**a**–**c**), (**d**–**f**), and (**g**–**i**) correspond to $\langle\Omega_W\rangle/\langle\Omega_S\rangle$ = 0.097, 0.23, and 0.45, respectively. The figures represent typical imaging (time of flight is 8.5 ms) and the corresponding column density with the best fit bimodal function (**a**, **d**, **g**), relative population (**b**, **e**, **h**), and the site distribution statistics for both total and condensate atom numbers (**c**, **f**, **i**), as introduced in Fig. 2. BEC fractions are **a** 0.116(1), **d** 0.075(1), **g** 0.076(1), respectively. Compared to the 3-site system which mainly populates the lower hyperfine levels with negligible inelastic loss, the 5-site system mainly populates the upper hyperfine levels which suffer from larger inelastic collisions[67–70] with more complex microwave couplings, making

evaporative cooling more challenging. The panel titled Theory (Edge, $\delta = 0$) shows a site distribution of the edge state calculated from diagonalizing the Hamiltonian in the Eq. (3) with $\delta = 0$. One can see a relatively large mismatch between the theoretical prediction and the data in the case of the largest microwave power, beyond the regime of $\Omega_S^{(1,2)} > \Omega_W^{(1,2)}$. Since this case corresponds to the largest change of populations, and thus possibly suffers relatively much severely from imperfections such as magnetic field fluctuation. The blue bars in the Exp.(Total) (Exp.(BEC)) in **c**, **f**, **i** show $(N_1 + N_3 + N_5)/N$ $((N_{c,1} + N_{c,3} + N_{c,5})/N_c)$. The orange bars in the Exp.(Total) (Exp.(BEC)) show $(N_2 + N_4)/N$ $((N_{c,2} + N_{c,4})/N_c)$. Source data are provided as a Source Data file.

coherence persisted throughout the entire process; the system keeps adiabaticity even in the presence of random fluctuations in $\delta$.

We can estimate a quantity $I = |\langle\Psi|\psi_{final}\rangle|$ from the relative population of site 5 in the final state $|\psi_{final}\rangle$, because ideally other sites are not occupied. The results shown in Fig. 5 indicate the quantity of $I=0.90(1)$ for the condensates, consistent with 0.82(2) in terms of population, shown in Fig. 5c. This value of $I$ close to 1 indicates that we successfully produce condensates in the edge states by gain engineering technique.

## Discussions

Despite the fact that the topological edge state should have no occupation in sites 2 and 4, the experiment observed small occupation; this occupation at site 2 and site 4 must be due to chiral-symmetry breaking terms in the Hamiltonian, which in our case is a magnetic field fluctuation $\delta$. Large fluctuations seen in the site occupation of BEC comes from the small BEC fraction: for the 5-site experiment the state mainly populates the upper hyperfine levels which suffer from inelastic collisions.

We also performed the numerical simulation in the presence of a static detuning due to the slow drift of a magnetic field, which breaks

the chiral symmetry for the 5 site system and dark-state condition for the 3-site system. As expected, we found the finite populations in the state 3 and even sites in the case of 3- and 5-site systems, respectively, at the end of the partial STIRAP procedure and before evaporation.

One of the characteristics of topological systems is robustness to external disturbances. Indeed, our topological atom laser is realized despite the natural inhomogeneity or disorder in the coupling, $\Omega_S^{(1)} \neq \Omega_S^{(2)}$ and $\Omega_W^{(1)} \neq \Omega_W^{(2)}$ (See "Methods"). We should also note that the zero-energy topological edge state is not robust against onsite disorders, namely again nonzero $\delta$. The existence of nonzero population in sites 2 and 4 seen in the experiment is ascribed to nonzero $\delta$. Even though the onsite disorder makes the energy of the edge state to become nonzero, the edge localized mode can still persist as long as the effect of $\delta$ is smaller than the band gap, $2(\Omega_S-\Omega_W)$, which is the case in our experiment.

What happens if the initial thermal atoms are not prepared only in the edge state, like the photonic systems, is an interesting problem. Our simulation where the initial state is not the pure edge state shows, although the fidelity does not reach 1 at $t = 0$, upon introducing gain for $t > 0$, the BEC emerges with the fidelity approaching 1, surpassing that of the thermal state. This is similar to the observation in the photonic

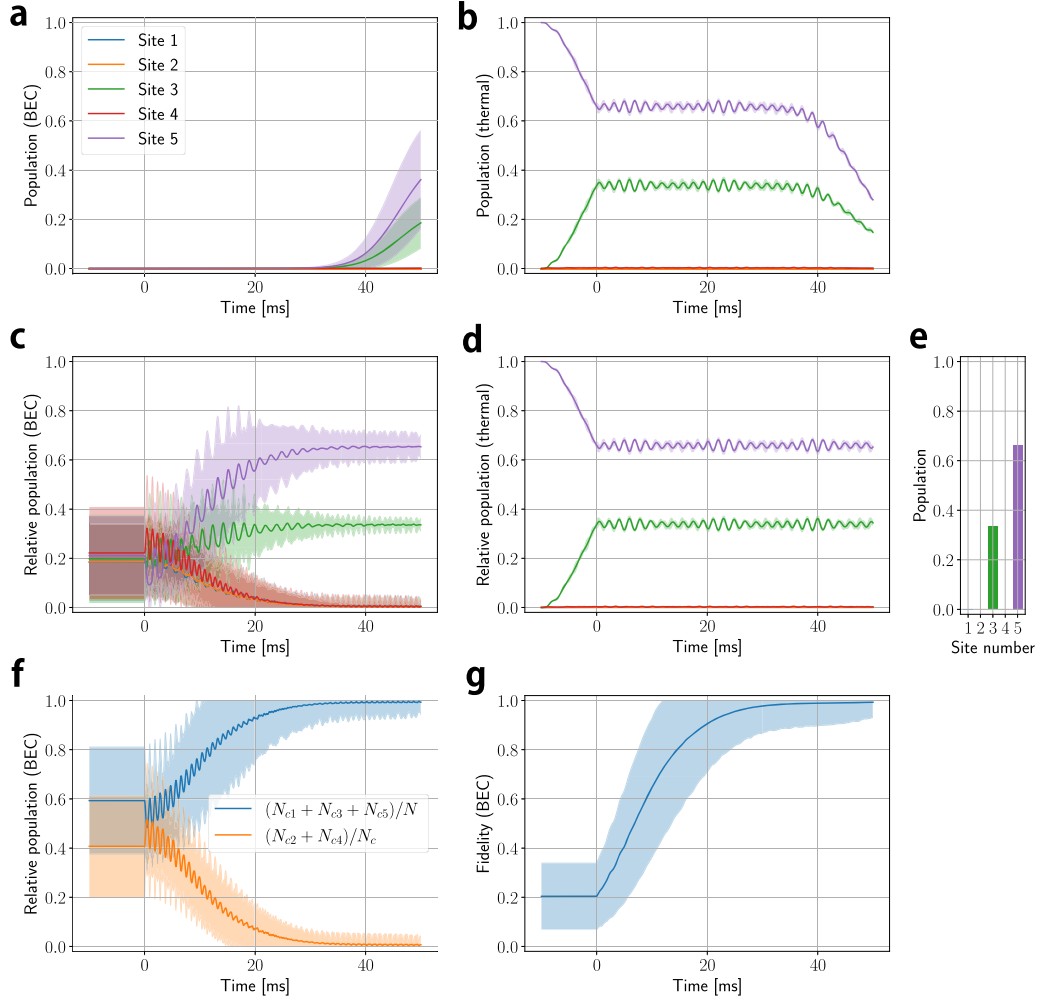

**Fig. 4 | Numerical results of the 5-site model with $\langle\Omega_W\rangle/\langle\Omega_S\rangle = 0.23$.** Time evolution of population of **a** condensate and **b** thermal atoms. $t = 0$ corresponds to the time at which gain is turned on. Relative population of **c** condensation and **d** thermal atoms so that the sum of the relative population is 1 at each time. **e** Expected edge state population obtained by diagonalization of the Hamiltonian of Eq. (3) in the main text with $\delta = 0$. **f** Time evolution of $(N_{c,1} + N_{c,3} + N_{c,5})/N_c$ and $(N_{c,2} + N_{c,4})/N_c$. **g** Time dependence of fidelity $|\langle\psi(t)|\Psi\rangle|/\sqrt{|\langle\psi(t)|\psi(t)\rangle\langle\Psi|\Psi\rangle|}$ where $|\psi(t)\rangle$ is obtained from numerical calculation, $|\Psi\rangle$ is the expected edge states shown in (**e**). Solid lines in **a**–**d**, **f**, **g** show the averages of the results computed from 100 different randomly-chosen initial values, and shaded areas indicate their standard deviation. Source data are provided as a Source Data file.

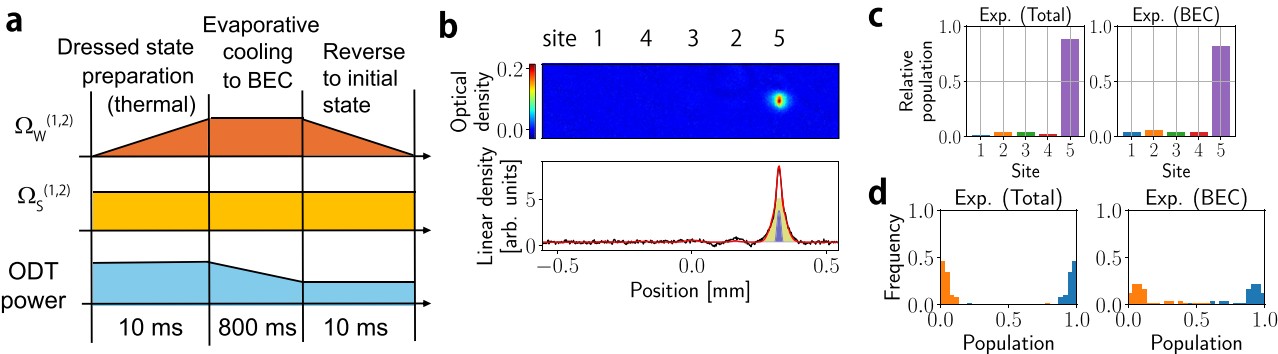

**Fig. 5 | Confirmation of coherence of superposition on the BEC in the edge state. a** Experimental sequence for 5-site reverse experiments. **b** The data taken at $\langle\Omega_W\rangle/\langle\Omega_S\rangle = 0$ after reversed process from $\langle\Omega_W\rangle/\langle\Omega_S\rangle = 0.45$ are shown. The figures **b** correspond to typical imaging (time of flight is 8.5 ms), the corresponding column density with the best fit bimodal function, **c** relative population, and **d** the site distribution statistics for both total and condensate atom numbers, as introduced in Fig. 3. BEC fraction is 0.166(6). Source data are provided as a Source Data file.

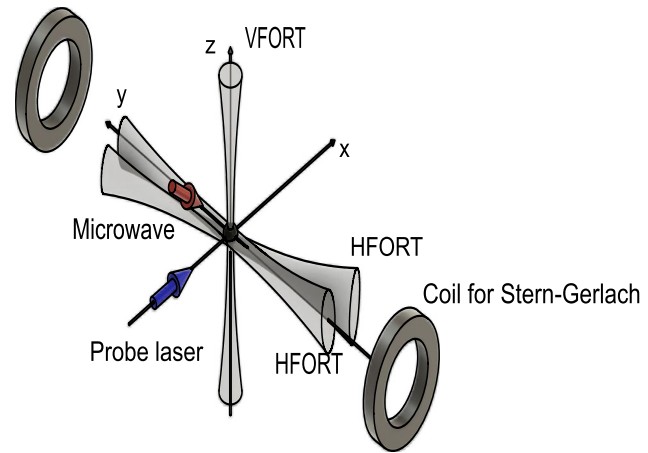

**Fig. 6 | Experimental setup.** The configurations of the optical trap, the microwave irradiation and probe axis as well as coils for Stern–Gerlach separation are shown. The bias magnetic field is applied so that the quantization axis is the y axis.

systems. In our actual experiments, however, significant magnetic field fluctuations were present, preventing the observation of behaviors predicted by numerical calculations.

Note that non-Hermitian dynamics can be realized through the demonstrated gain engineering and atom loss. For example, if we prepare thermal atoms populated at sites 1, 3, and 5, and introduce dissipation at sites 2 and 4, we should be able to realize a $\mathcal{PT}$-symmetric non-Hermitian system. Such models should feature $\mathcal{PT}$-symmetric/broken phase transition, which should be observable within current experimental techniques.

Here, the subsystem which can be described in terms of the non-Hermitian Hamiltonian is the Bose–Einstein condensate part of the atoms. The external bath which acts to provide the gain is the thermal (non-condensed) atoms. By adding spin-dependent gain and loss in the spin-orbit coupled ultracold gases[44,63], for example, one should also have the non-Hermitian skin effect, characteristic of a non-Hermitian system.

By preparing the initial thermal atoms in a proper dressed state of the synthetic lattice, we observe the formation of BEC in the dressed state even when the dressed state is not the ground state of the lattice. Such a formation of BEC in a dressed mode can be regarded as laser oscillation of an atomic matter wave. In particular, the 5-site experiment demonstrates topological atom laser oscillating at the topological edge state. The current limitation is the external field fluctuations which can be improved by appropriate shielding and active stabilization[64–66]. Increasing Rabi frequencies and shortening the experimental time scale will help reducing the effect of both field fluctuations and inelastic loss.

Note that we believe there is nothing special about topological atom laser in terms of characteristics such as threshold behavior, spectral properties, and the second-order coherence compared to the conventional BEC, although they are subjects for future research.

Since there is no BEC for fermions, our work cannot be directly applied to ultracold fermions. Lasing is a phenomenon specific to bosons. Gain engineering should, however, work for fermionic condensates like Bardeen–Cooper–Schriefer or molecular BEC states.

Our results successfully demonstrate that an effective state-dependent gain can be engineered through evaporative cooling by appropriate preparation of the initial thermal state. Such a gain-loss control in ultracold atomic gases provide a versatile tool to study non-Hermitian physics. Combining with the ability to control losses and inter-particle interactions, one can now

explore much broader classes of non-Hermitian quantum many-body physics in ultracold atomic gases. Our results also show that atom optics can be combined with concepts developed in topological photonics, opening the field of topological atom optics. While the main technological implication of our work is the ability to introduce gain in ultracold gases, which allows one to explore a variety of non-Hermitian Hamiltonians in ultracold settings, the topological atom laser should share the merits the topological photonic lasers have, such as the robust single-mode lasing (condensation) in the presence of disorder. This work thus serves as the next footstep to further studies and applications of topological and non-Hermitian physics.

## Methods
### Experimental setup
Our experiments start with $^{87}$Rb atoms above the critical temperature for BEC trapped in the crossed optical dipole trap at about 1064 nm (see Fig. 6). The crossed optical dipole trap is created with two horizontal beams which are crossing at slight angle to provide a large trap volume and a vertical beam (see Fig. 6). Then the $^{87}$Rb atoms are pumped to a hyperfine state $|F=1, m_F=1\rangle$ of the $^2S_{1/2}$ electronic ground state by optical pumping. For 5-site experiment, we further transfer atoms into the $|F=2, m_F=2\rangle$ state by ARP with microwave irradiation. For efficient evaporative cooling, we turn off a deeper horizontal beam in the middle of evaporative cooling. In order to induce coupling between the hyperfine states, or neighboring sites in the synthetic dimension, we apply resonant microwave fields. The subsequent sequence of the microwave irradiation consists of two stages: First, we apply time-dependent microwaves to prepare thermal atoms in a desired superposition of the synthetic lattice sites. Then, further evaporative cooling is performed with microwave couplings kept constant. Resulting formation of a BEC can be regarded as lasing of atoms at a specific eigenmode in the lattice.

### Generation of 4-color microwave
We simultaneously apply 4-color microwaves around the clock frequency ~6.8 GHz. We use an output from a microwave synthesizer at a frequency $f$ for the $|2, -2\rangle - |1, -1\rangle$ coupling and generate three sidebands. An image rejection mixer can create a microwave sideband only at higher frequency side at a frequency of $f + f_{RF1}$, and is set on resonance to the $|2, 0\rangle - |1, 1\rangle$ transition. A series connection of an additional image rejection mixer further generates sidebands at $f + f_{RF2}$ and $f + f_{RF1} + f_{RF2}$, and are set to the $|2, 0\rangle - |1, -1\rangle$ and $|2, 2\rangle - |1, 1\rangle$ resonances (Fig. 7). In this configuration, $\Omega_W$ can be controlled by the strength of the second RF. Measured Rabi frequencies for the 3-site and the 5-site experiment are shown in Tables 1 and 2, respectively. Numerical solutions to the 5-site SSH Hamiltonian using the measured Rabi frequencies are shown in Fig. 8.

### Suppression of magnetic field fluctuation
One of the major sources of the short-term magnetic field fluctuations comes from electric devices around the experimental chamber. Transducers in power supply emit strong noises at alternate current (AC) line frequency (60 Hz) and its harmonics which amounts to ~0.8 mG at the chamber. We construct a feed-forward system to suppress the AC field fluctuation. We monitor field fluctuations by a flux gate sensor placed near the magnetic coil on the quantization axis. A function generator with a reference clock of 10 MHz locked to the AC line (DS technology) generates 60 Hz harmonics up to 7th order to be added to the analog control of the coil. The amplitudes and phases of the harmonics is tuned to minimize the fluctuation of sensor output. This enables us to suppress the 60 Hz line noise down to ~0.1 mG at all times during the experiment.

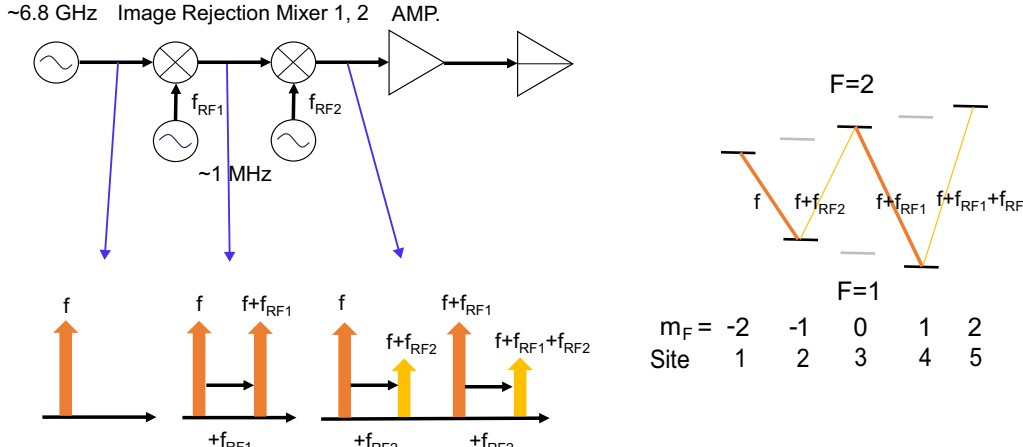

**Fig. 7 | Schematic of the microwave setup.** Two RFs are mixed to the main microwave by image rejection mixers (left). The frequency components at several stages are also shown. The resulting microwave has four frequency components set on-resonant to the transition involved in the 5-site experiment, as shown in the ground state energy diagram of $^{87}$Rb (right). In this configuration, $\Omega_W$ can be controlled by the strength of RF2.

## Table 1 | Rabi frequencies for the 3-site experiment

| $\Omega_1/\Omega_2$ | $T_{prep}$ | site 2 - 3 ($\Omega_1/2\pi$) | site 3 - 4 ($\Omega_2/2\pi$) |
|---|---|---|---|
| 19.27 | 3 ms | 1.457 kHz | 0.07560 kHz |
| 1.345 | 5 ms | 0.8410 kHz | 0.6252 kHz |
| 0.02870 | 8 ms | 0.03283 kHz | 1.144 kHz |

## Table 2 | Rabi frequencies for the 5-site experiment

| $\langle\Omega_W\rangle/\langle\Omega_S\rangle$ | site 1 - 2 ($\Omega_S^{(1)}/2\pi$) | site 3 - 4 ($\Omega_S^{(2)}/2\pi$) | site 2 - 3 ($\Omega_W^{(1)}/2\pi$) | site 4 - 5 ($\Omega_W^{(2)}/2\pi$) |
|---|---|---|---|---|
| 0.09691 | 2.119 kHz | 0.8646 kHz | 0.07464 kHz | 0.1319 kHz |
| 0.1560 | 2.116 kHz | 0.8605 kHz | 0.1081 kHz | 0.3847 kHz |
| 0.2331 | 2.120 kHz | 0.8625 kHz | 0.1379 kHz | 0.6114 kHz |
| 0.3733 | 2.123 kHz | 1.363 kHz | 0.2387 kHz | 0.8563 kHz |
| 0.4466 | 3.092 kHz | 0.8446 kHz | 0.3787 kHz | 1.424 kHz |

## Numerical simulation: Hermitian term

The Hermitian part of the Hamiltonian (i.e., without gain and loss) for our numerical simulation is time-dependent. Similar to Eq. (3) in the main text, we employ the following Hamiltonian

$$\hat{H}_0 = \frac{\hbar}{2}\begin{pmatrix} -4\delta(t) & \Omega_{12}(t) & 0 & 0 & 0 \\ \Omega_{12}(t) & 2\delta(t) & \Omega_{23}(t) & 0 & 0 \\ 0 & \Omega_{23}(t) & 0 & \Omega_{34}(t) & 0 \\ 0 & 0 & \Omega_{34}(t) & -2\delta(t) & \Omega_{45}(t) \\ 0 & 0 & 0 & \Omega_{45}(t) & 4\delta(t) \end{pmatrix} \quad (5)$$

where $\Omega_{i,i+1}(t)$ ($i = 1, 2, 3, 4$) is the Rabi frequencies in microwave couplings between site $i$ and site $i + 1$, and $\delta(t)$ is the detuning due to a magnetic field fluctuation. We only consider the magnetic filed fluctuation whose frequency is 60 Hz, namely,

$$\delta(t) = \delta_0 \cos(2\pi f_0 t + \varphi_0), \quad (6)$$

where $\delta_0$ is a strength, $f_0$ is the frequency of commercial power supply (60 Hz) and $\varphi_0$ is a phase offset. The phase offset is chosen randomly. In our numerical analysis described below, averaging over many samples is performed, so the effect of this random selection of phase offsets does not remain.

## Numerical simulation: gain term

The thermal atoms present in the system provide the gain through the evaporative cooling. In order to model this process, we assume that the thermal atoms are kinetically thermal but made of coherent superposition of spin states. We denote the (coherent) spin components of the thermal atoms as $|T\rangle$, which we take to be normalized. The strength of the gain is proportional to the number of atoms in the thermal components, which we denote by $n_T$. We note that $n_T$ is a time-dependent quantity. The gain is then modeled by including the following term in the Hamiltonian

$$\hat{H}^{gain}(n_T) \equiv i\hbar g n_T |T\rangle\langle T|, \quad (7)$$

with the coefficient $g$ being the effective strength of the gain per unit thermal component. We note that this Hamiltonian, which is a function of the thermal atom density $n_T$, acts on the condensate wavefunction $|\psi(t)\rangle$ and increases the condensate density $\langle\psi(t)|\psi(t)\rangle$ but does not directly change the number of thermal atoms $n_T$. As we discuss later, we decrease $n_T$ in an ad hoc manner to account for the increase of the condensate density.

The spin component $|T\rangle$ of the thermal atoms we use in our simulation is determined by following realistic simulation of state preparation in the actual experiment. We start from the initial condition that all of the population is in site 5. Then, we simulate the time evolution under a linear increase of the Rabi coupling strength from zero to the final value $\Omega_{j,j+1,final}$ as described in the state initialization part of Fig. 1:

$$\Omega_{j,j+1}(t) = \left[\frac{t+10}{10}\right]\Omega_{j,j+1,final} \quad (8)$$

where $t$ is time in millisecond. We note that we change from $t = -10$ to 0. The state realized at $t = 0$ is what we use for $|T\rangle$.

## Numerical simulation: loss term

Two-body inelastic collisions are the dominant source of atom loss of Bose condensates in our system. Two-body inelastic loss of condensate $^{87}$Rb have been studied[67–70]. For simplicity, we set the two-body loss coefficient $\gamma$ to be $\gamma = 1 \times 10^{-13}$ cm$^3/s$ for the atom pairs of

- ($F = 2, m_F = 2$) and ($F = 2, m_F = 0$)
- ($F = 2, m_F = 2$) and ($F = 2, m_F = -2$)

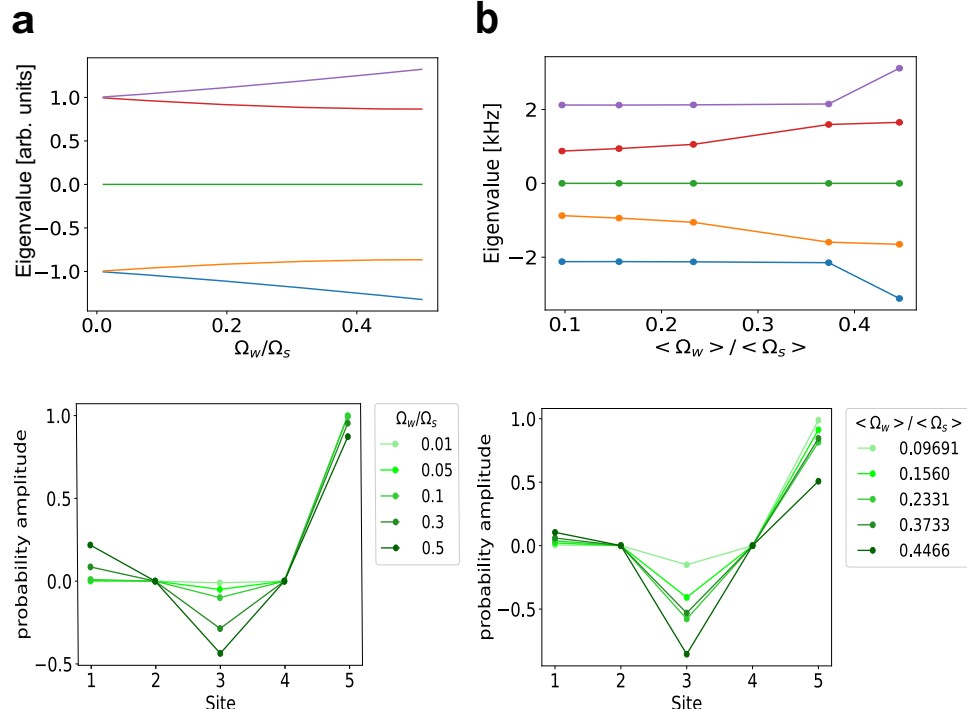

**Fig. 8 | Numerical solutions to the 5-site SSH Hamiltonian. a** Solutions to the 5-site SSH Hamiltonian (Eq. (5)) with $\Omega_i^{(1)} = \Omega_i^{(2)} = \Omega_i$ ($i$ = S, W) and $\delta = 0$. (top) Dependence of the eigenenergies on the ratio $\Omega_W/\Omega_S$ and (bottom) eigenvectors for several Rabi frequency ratio are plotted. **b** Solutions to the experimentally realistic

Hamiltonian. Imbalance between $\Omega_i^{(1)}$ and $\Omega_i^{(2)}$ are considered and the dependence on their averaged values are shown. Characteristic features of the edge state, zero-energy and vanishing amplitude on the site 2 and 4, are retained.

- ($F = 2$, $m_F = 0$) and ($F = 2$, $m_F = -2$)

We take the loss coefficient to be $2\gamma$ for the pair of
- ($F = 2$, $m_F = 0$) and ($F = 2$, $m_F = 0$)

and zero for other combinations.

This is equivalent to considering the following (nonlinear) Hamiltonian

$$\hat{H}^{\text{loss}} \equiv -i\hbar\gamma \begin{pmatrix} n_3 + n_5 & 0 & 0 & 0 & 0 \\ 0 & 0 & 0 & 0 & 0 \\ 0 & 0 & n_1 + 2n_3 + n_5 & 0 & 0 \\ 0 & 0 & 0 & 0 & 0 \\ 0 & 0 & 0 & 0 & n_1 + n_3 \end{pmatrix}, \quad (9)$$

where $n_i \equiv |\psi_i|^2$ is the condensate density of $i$th site. The choice of this specific form of loss Hamiltonian of Eq. (9) is based on the reported measurement results, and does not have a direct relation to the appearance or suppression of the populations in the even sites where the loss is set to zero. In addition, we use the typical atom density of our experimental setup as $10^{14}$cm$^{-3}$.

**Numerical simulation: procedure**

We start from the random initial state $|\psi\rangle$ with its norm equals to $10^{-6}$. This random initial seed of condensate is necessary to obtain the growth of condensate density in our numerical simulation. We then time evolve according to the nonlinear and non-Hermitian Schrödinger equation

$$i\hbar\frac{d}{dt}|\psi\rangle = \left(\hat{H}_0 + \hat{H}^{\text{gain}}(n_T) + \hat{H}^{\text{loss}}\right)|\psi\rangle. \quad (10)$$

In our numerical simulation, we discretize the time with the step of $\Delta t$ and calculate

$$|\psi(t + \Delta t)\rangle = e^{-i\hat{H}^{\text{loss}}\Delta t/\hbar} e^{-i\hat{H}^{\text{gain}}(n_T)\Delta t/\hbar} e^{-i\hat{H}_0\Delta t/\hbar}|\psi(t)\rangle. \quad (11)$$

After every step we calculate the multiplication by $e^{-i\hat{H}^{\text{gain}}(n_T)\Delta t/\hbar}$, we evaluate how much the norm of the wavefunction $\langle\psi(t)|\psi(t)\rangle$ is increased, and subtract the same amount from $n_T$ to update the value of $n_T$.

**Numerical simulation: results of 5-site model**

To perform numerical simulation, we need to determine the coefficients of gain $g$, loss $\gamma$, and detuning $\delta$. For the gain, we use $g = 150$ Hz in order to ensure good agreement between experiment and numerical calculation. The value for loss was set at 10 Hz based on typical atomic density of $10^{14}$ /cm$^3$ and inelastic collision loss rate of $1 \times 10^{-13}$ cm$^3$/s. For the detuning, since our magnetic field fluctuations are about 0.1 mG, we used the corresponding value.

Since we also want to take into account the adiabaticity of the state preparation, we first prepare the initial state with the thermal atoms at site 5 only (see Fig. S1 in the Supplementary Information). In the numerical simulation, the initial state of the condensate and phase of magnetic field fluctuation contain random factors. Therefore, we average the results of about 100 simulations.

## Data availability
The data supporting the results presented in this paper are provided with this paper as Source data files.

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

## Acknowledgements
We thank to K. Shibata and S. Tojo for discussion on inelastic collisions in Rb. This work was supported by the Grant-in-Aid for Scientific Research of JSPS (No. JP17H06138 (Y.Takahashi), No. JP18H05405 (Y.Takahashi), No. JP18H05228 (Y.Takahashi), No. JP21H01007 (T.O.), No. JP21H01014 (Y.Takasu) and JP24K00548(T.O.)), JST CREST (Nos. JPMJCR1673 (Y.Takahashi), No.JPMJCR19T1(T.O.) and JPMJCR23I3 (Y.Takahashi)), JST PRESTO Grant No. JPMJPR2353 (T.O.), MEXT Quantum Leap Flagship Program (MEXT Q-LEAP) Grant No. JPMXS0118069021 (Y.Takahashi), JST Moonshot (R&D Grant Nos. JPMJMS2268 (Y.Takahashi) and JPMJMS2269 (Y.Takahashi)), and JST ASPIRE (No. JPMJAP24C2 (Y.Takahashi)).

## Author contributions
T.T. and S.T. contributed equally to this work. T.T. and S.T. carried out experiments. T.T. and Y. Takasu analyzed the data. T.O. gave advice on experiments from the theoretical viewpoint. Y. Takasu and T.O. performed the numerical calculation. K.Y. contributed in the early stage of the experiment. Y. Takahashi conducted the whole experiment. All the authors contributed in discussing the result and writing the manuscript.

## Competing interests
The authors declare no competing interests.
