## [Transparent Peer Review file · Nature Communications]

Gain engineering and atom lasing in a topological edge state in synthetic dimensions

Corresponding Author: Dr Yosuke Takasu

Version 0:

Reviewer comments:

Reviewer #1

(Remarks to the Author)

In the manuscript entitled "Gain engineering and topological atom laser in synthetic dimensions", authors consider a multi-level bosonic Rb-87 system in which quantum states can be coherently controlled through two-photon processes. In particular, two hyperfine manifolds ($F=1,2$) are chosen to introduce 3 or 5 synthetic sites coupled by microwave pulses. In this well-controlled platform, authors experimentally examine how thermal atoms cool down into different states – different sites in the synthetic dimension manner – when evaporative cooling is introduced. A dark state with the STRIP sequence – which is a well-known technique in quantum optics/AMO – and the SSH model with alternating hopping strengths are realized in this system.

For both models, authors observe the emergence of Bose-Einstein condensation at a specific site – usually at the edge – and this is interpreted as the lasing mode induced by gain engineering. This is of particular importance for pushing the limit of non-Hermitian dynamics, which is one of the frontier directions at the moment. The Authors point out that this gain engineering remains challenging in quantum systems in contrast to photonics, which is indeed true.

Overall, the performed experiments seem very interesting with rock solid data sets. As noted, it is nice to see how topological edge mode (or a dark state) allow the control of thermal population. To me, the "observation" appears interesting on its own.

I have some concerns about interpreting the experimental observations in terms of "gain engineering." While the introduction connects gain engineering with non-Hermitian dynamics, this connection needs more clarity. In the presented experiment, the gain term is site-specific. Could the authors provide examples of how non-Hermitian dynamics can be realized through the demonstrated gain engineering and atom loss? It's not immediately clear how the proposed gain engineering helps implement a genuine PT-symmetric Hamiltonian with balanced gain-loss.

Additionally, what is the sub-system being considered when envisioning a non-Hermitian Hamiltonian? Since the authors discuss non-Hermitian (or open quantum) systems, readers would benefit from understanding how atoms flow into the subsystem as a gain term. Can the 3-site (or 5-site) system be described by non-Hermitian Hamiltonian? Would the non-Hermitian skin effect be observable with this gain term? I feel the data set presented here seems interesting, but it would be beneficial if authors put this in the right context.

Another question author may clarify is the quantum statistics. Like photons in photonics, it is not surprising that gain engineering properly works for bosonic particles, especially when BEC emerges, thanks to the Bose enhancement. Can authors comment how proposed gain engineering works for fermions?

In summary, while I have several concerns and suggestions outlined above that the authors should address before publication in Nature Communications, I believe this work demonstrates significant breakthroughs in multiple areas—including quantum system gain engineering, topological lasing with bosonic atoms, and potential developments in non-Hermitian engineering. I look forward to reviewing the revised manuscript before making my final recommendation.

Reviewer #2

(Remarks to the Author)

The authors realise Bose-Einstein condensation in a micro-wave dressing of the hyperfine levels of ^{87}Rb atoms. They perform evaporative cooling in the dressed states after carefully preparing the initial thermal states. In this approach, the thermal atoms can be seen as a reservoir, which provides “gain” for a “laser” formed of condensate atoms. They demonstrate this experimental approach in two configurations with 3 sites and 5 sites respectively, where they engineer a SSH model, one of the simplest topological models in 1d. There, condensation occurs in a dressed state, which is a topological edge state of the 5-site system. Condensation is observed and quantified by observing the momentum distribution in time-of-flight with a magnetic gradient ; fitting the bimodal distribution identifies the condensed atoms and the thermal atoms. A novel aspect is the ability to engineer gain in a setup with ultracold atoms, where it is usually difficult to obtain, contrary to losses, which opens a new path to study non-Hermitian physics with both gain and losses. A theoretical analysis is provided to support the claims and the interpretation of the experimental data, which is based on solving the non-Hermitian Schrödinger equation for the condensate wavefunction, taking the thermal atoms as a gain medium (which empties as the condensate grows), in the presence of losses for the condensate.

Overall I appreciated the manuscript and the work, but I have several questions and remarks for the authors, that would need to be addressed before supporting publication :

1) I find that the denomination “topological atom laser” is a bit overselling and could be simplified to : Bose-Einstein condensation / Lasing in a topological edge state.

2) While similar experiments have been performed in photonic systems, such as in lattices of polariton micropillars realising a SSH model [ref 9 of the authors], this is the first observation in cold atoms. In the polariton experiment, the system is pumped incoherently and above a critical threshold, lasing suddenly dominates in an edge mode. However, here, the authors explain lasing in the edge state on page 2, right column, by a careful preparation of the reservoir of thermal atoms. As the atoms are initially in the dark state due to the time-dependent micro-waves, the population in the other states is indeed negligible, and thus we both only expect the dark state to be populated and the dark state to have a significant condensate fraction. This is obvious if there is no mechanism that would connect atoms in the dark state to a lower-energy state. In my opinion, this differs from the lasing demonstrated in other systems, where all modes are pumped and one wins above a critical threshold ; here, only the dark state is occupied by the thermal atoms which naturally forces Bose-Einstein condensation in this mode.

From the sentence “We see that in all three values of r , the BEC phase transition, namely, atom lasing is successfully observed, and the formed BEC has the largest population in site 5, with certain atom numbers in 3 and 1 with almost no occupation in 2 and 4, which is consistent with the wavefunction profile of the topological edge state of the SSH model.”, I do not see what else I should expect.

It would be interesting to explain what would happen if the thermal atoms are prepared in a state which is not dark for example and discuss the robustness of the scheme with the preparation of the thermal atoms in a particular state. A clearer explanation of the role of the dark state is necessary for me.

3) The most natural description of such a system for me is the one of a spinor condensate. In the dressed picture, the eigenstates are decomposed over the hyperfine states which can interact at low temperature with s-wave collisions with interaction parameters that depend on the magnetic field and on the states. I did not see any discussion or comment on interactions parameters, which I would appreciate.

4) Related to my 2nd comment, I would find instructive to see what are the populations in the different hyperfine states at the end of the partial STIRAP procedure and before evaporation, to be able to compare them with the observed populations after evaporation. For example, in figure 2b, I would like to understand if the population in the state 3 originates purely from imperfections in the STIRAP procedure. This could be compared to the results of the theoretical model and figure 4d-e, which indicate that “that the edge state are adiabatically prepared”.

5) In the 5-site SSH configuration, the condensate fraction are much smaller than in the 3-site configuration. Why is that ? The total condensate fraction that is reached is not written in any case.

6) On page 5, equation 4 : the coefficient g is not explained. How can it be deduced from a microscopic analysis ?

7) On page 6, figure 4g, can the fidelity be accurately extracted at times $t < 30\text{ms}$, where the population in the BEC state is actually very close to zero ? It seems to me that the fidelity is above 0.95 as soon as the population is non negligible.

Related to my 4th comment, what is the expected fidelity for the thermal component, especially at $t=0$ at the end of the MW preparation ? What is the sensitivity of the lasing in the dark state on the MW preparation based on the theoretical model ?

8) On page 7, the part on “Confirming the coherence of the topological atom laser”, along with figure 5, is important in my opinion and I appreciated it. How is the fidelity 0.95(5) estimated ? It is not obvious for me based on the data of figure 5 and the relative population in state 5 below 0.8.

Additional smaller comments :

1) In the figures 2 and 3, are the fourths rows commented in the main text ? I did not see their use. It seems to me that the sentence beginning by “In the third (fourth) row of Fig. 2,” on page 2, right column, does not exactly matches the content of the figure.

2) On page 3, right column : typo “As we see in in Methods”

3) On page 3, equation 3 : the indices (1,2) are not explicited for the Rabi frequencies in microwave coupling

- 4) On page 4, right column, the reasons for the “not a simple ascending order” could be explicitly explained in a short way, based on the difference in sign of the magnetic moments of states of same projection m_F between the $F=1$ and $F=2$ states.
- 5) The sentence with the definition of H_{gain} on page 5 left column does not have a verb.

Reviewer #3

(Remarks to the Author)

Summary

The authors report the experimental observation of BEC formation in excited eigenstates of a synthetic lattice of hyperfine states of ^{87}Rb atoms. Their gain engineering scheme is original and well described, and supported by a theoretical model. They further employ their scheme to engineer a 5-site synthetic SSH chain, in which they promote lasing from topological edge modes. The technique presented in the article can catalyze the exploration of driven-dissipative phenomena in ultracold atomic gases.

General Remarks

The article is well detailed, both in terms of the experimental protocol and the theoretical modeling and simulations. I think that, overall, it warrants publication in Nature Communications. I do have, however, several remarks that I would like the authors to address.

1. The information in the article is quite scattered throughout the pages, and early pages and statements become clear only later on. The reader, therefore, has to hold on to several bits of information before they are eventually clarified, which in my opinion leads to a confusing reading experience. There are also several repetitions, for example the abstract and the first lines of the introduction, or in the “Discussions” section.
2. What is the rationale behind making an atom laser topological, especially an SSH (or 1D, in general) one? What would be the advantages (and disadvantages) over a conventional atom laser?
3. Also in connection with my previous question, the authors show BEC formation, but what about lasing characteristics? What’s the threshold behavior? What are the spectral properties? What’s the second order coherence?
4. This is more of a curiosity, not a request for review; would it be possible to directly measure the Zak phase in this particular synthetic lattice?

Specific Remarks

1. Just before eq. 1, the authors write “ λ -type”, although later they use “ Λ -type” and “W-type”. Please use a consistent notation.
2. At the beginning of the right column of page 2, they authors write “ τ/τ_{full} leads to the equal superposition of site 4 and 2”. While this seems to be clear from their previous expression for the dark state (at $\delta=0$) and from the experimental results of Fig. 2b, third row, the “Theory” panel on the same row seems to predict a non-symmetric distribution. Why is that? Does it also account for a non-zero δ ? And if so, why the population on site 3 is still zero?
3. Again in Fig. 2, the legend text in the bottom row is way too small, and the remaining text could also be made a little bigger to improve legibility. If necessary, the authors can make more space by collapsing repeated x- and y-labels. This also applies to Fig. 3 and 1.
4. Continuing the discussion of Fig. 2, why the a) and c) columns are show for $\tau/\tau_{\text{full}} = 0.3$ and 0.8 ? What happens at $\tau/\tau_{\text{full}} = 0$ and 1 ? Aren’t the “Theory” distributions predicted for those values?
5. Let’s go back to Fig. 1. Why the site ordering 1-4-3-2-5 in column c? In the same column, are Ω_S and Ω_W the averages defined later in the text? Shouldn’t the 5-site frequencies in the left panel be marked as $\Omega_S^{(1,2)}$ and $\Omega_W^{(1,2)}$?
6. Why is there a large mismatch wrt the theoretical prediction in the rightmost panel of the third row in Fig. 3?
7. In the “Discussions” section, it’s stated again that a small occupation in sites 2 and 4 is to be expected due to the non-zero magnetic field fluctuations δ that break the chiral symmetry. While this seems to be incorporated in the theory, the “Theory” panels throughout the text have identically zero occupation in the even sites. Why is that?
8. In relation to my previous answer, why do the authors choose the loss coefficient they way they do just before equation 9? There doesn’t seem to be any justification for their choice. This leads to the loss Hamiltonian in equation 9, which is 0 on sites 2 and 4. Could this be the source of the identically zero population on sites 2 and 4 in the “Theory” panels throughout the text? Wouldn’t this choice make the non-zero condensate density solution in sites 2 and 4 locally (dynamically) unstable, and thus forbidden in the numerical solutions?
9. The legends in the last row of Fig 5 are again too small.

Version 1:

Reviewer comments:

Reviewer #1

(Remarks to the Author)

I have thoroughly reviewed the revised manuscript with considerable interest and find that the authors have addressed all

referee comments and suggestions comprehensively and thoughtfully. The authors have not only implemented specific technical corrections but have also substantially improved the clarity throughout the paper. Particularly noteworthy is how they have addressed the common concerns raised by multiple referees, providing additional data, analyses, and explanations that significantly strengthen their conclusions.

This innovative work represents a significant advancement in the quantum simulation field, with potential implications for both theoretical understanding and practical applications. The data presentation is clear and accessible.

Based on my assessment, this manuscript fully satisfies the criteria of Nature Communications for novelty, significance, and technical quality. I am therefore happy to strongly support its publication in its current form.

Reviewer #2

(Remarks to the Author)

The authors thoroughly answered my questions with additional numerical data to support their claims, as well as the questions from the other reviewers. In particular, they have clarified the role of the dark state and of the initial preparation of the thermal gas in the dark state.

They have clarified and improved the manuscript, and I do not have any concerns left and I advise publication.

Reviewer #3

(Remarks to the Author)

I thank the authors for their exhaustive reply; they have addressed all of my concerns. In particular, I find the new labeling with Ω_1/Ω_2 and explicit $\delta=0$ to be definitely clearer than before. I also appreciate the additional simulations prepared in response to Reviewer #2, which have also resolved some of my doubts.

As a result, I recommend the revised manuscript for publication.

Response to Reviewers

We thank all the reviewers for their careful reading of our manuscript and their positive feedback. Their comments were very useful to us, and their questions brought up interesting discussions with new insights. The reviewers raised several points that gain profit from further clarification. This has led to several improvements in the manuscript. Numbers of equations, figures, etc. are those in the revised manuscript unless otherwise stated. The summary of changes are included in the end of this document.

Reviewer #1 (Remarks to the Author):

In the manuscript entitled "Gain engineering and topological atom laser in synthetic dimensions", authors consider a multi-level bosonic Rb-87 system in which quantum states can be coherently controlled through two-photon processes. In particular, two hyperfine manifolds ($F=1,2$) are chosen to introduce 3 or 5 synthetic site coupled by microwave pulses. In this well-controlled platform, authors experimentally examine how thermal atoms cool down into different states – different sites in the synthetic dimension manner – when evaporative cooling is introduced. A dark state with the STRIAP sequence – which is well-known technique in quantum optics/AMO – and the SSH model with alternating hopping strengths are realized in this system.

For both models, authors observe the emergence of Bose-Einstein condensation at a specific site – usually at the edge – and this is interpreted as the lasing mode induced by gain engineering. This is of particular importance for pushing the limit of non-Hermitian dynamics, which is one of the frontier directions at the moment. The Authors point out that this gain engineering remains challenging in quantum systems in contrast to phonics, which is indeed true.

Overall, the performed experiments seem very interesting with rock solid data sets. As noted, it is nice to see how topological edge mode (or a dark state) allow the control of thermal population. To me, the "observation" appears interesting on its own.

I have some concerns about interpreting the experimental observations in terms of "gain engineering." While the introduction connects gain engineering with non-Hermitian dynamics, this connection needs more clarity. In the presented experiment, the gain term is

site-specific. Could the authors provide examples of how non-Hermitian dynamics can be realized through the demonstrated gain engineering and atom loss? It's not immediately clear how the proposed gain engineering helps implement a genuine PT-symmetric Hamiltonian with balanced gain-loss.

We appreciate your inquiries regarding gain engineering and non-Hermitian physics with a particular focus on PT-symmetric physics. As demonstrated in this study, the gain at each site was determined by the state of the thermal atoms. The loss arises from the intrinsic inelastic scattering between two atoms. In addition, various losses can be introduced artificially. Because we used an artificial dimension technique in this study, controlled *site-dependent* dissipation can be introduced by applying *spin-dependent* atomic loss. Thus, for example, if we prepare thermal atoms populated at sites 1, 3, and 5, and introduce dissipation at sites 2 and 4, we should be able to realize a PT-symmetric non-Hermitian system. Such models should feature PT-symmetric/broken phase transition, which should be observable within current experimental techniques. We have included information regarding the feasibility of realizing PT-symmetric non-Hermitian systems in the Discussion section.

The newly added sentence reads

“Note that non-Hermitian dynamics can be realized through the demonstrated gain engineering and atom loss. For example, if we prepare thermal atoms populated at sites 1, 3, and 5, and introduce dissipation at sites 2 and 4, we should be able to realize a PT-symmetric non-Hermitian system. Such models should feature PT-symmetric/broken phase transition, which should be observable within current experimental techniques.”

Additionally, what is the sub-system being considered when envisioning a non-Hermitian Hamiltonian? Since the authors discuss non-Hermitian (or open quantum) systems, readers would benefit from understanding how atoms flow into the subsystem as a gain term. Can the 3-site (or 5-site) system be described by non-Hermitian Hamiltonian? Would the non-Hermitian skin effect be observable with this gain term? I feel the data set presented here seems interesting, but it would be beneficial if authors put this in the right context.

We thank the reviewer for the question. Here, the sub-system which can be described in terms of the non-Hermitian Hamiltonian is the Bose-Einstein condensate part of the atoms. The “external” bath which acts to provide the gain is the thermal (non-condensed) atoms.

In order to observe the non-Hermitian skin effect, we need additional ingredients. For example, by adding spin-dependent gain and loss in the spin-orbit coupled ultracold gases [Ren et al., Nat. Phys. **18**, 385-389 (2022); Guo et al., Phys. Rev. A **106**, L061302 (2022)], one should have the non-Hermitian skin effect. We have added these discussions in the revised manuscript.

The newly added sentence reads

“Here, the subsystem which can be described in terms of the non-Hermitian Hamiltonian is the Bose-Einstein condensate part of the atoms. The “external” bath which acts to provide the gain is the thermal (non-condensed) atoms. By adding spin-dependent gain and loss in the spin-orbit coupled ultracold gases [Ren et al., Nat. Phys. **18**, 385-389 (2022); Guo et al., Phys. Rev. A **106**, L061302 (2022)], for example, one should also have the non-Hermitian skin effect, characteristic of a non-Hermitian system.”

Another question author may clarify is the quantum statistics. Like photons in photonics, it is not surprising that gain engineering properly works for bosonic particles, especially when BEC emerges, thanks to the bose enhancement. Can authors comment how proposed gain engineering works for fermions?

We thank the reviewer for the question. Since there is no clear difference between degenerate fermionic atoms and thermal atoms or, in other words, there is no Bose-Einstein condensation for fermions, our work cannot be directly applied to ultracold fermions. Lasing is a phenomenon specific to bosons. However, gain engineering should work for fermionic condensates like the BCS or molecular BEC states.

The newly added sentence reads

“Since there is no Bose-Einstein condensation for fermions, our work cannot be directly applied to ultracold fermions. Lasing is a phenomenon specific to bosons. Gain engineering

should, however, work for fermionic condensates like Bardeen-Cooper-Schrieffer or molecular BEC states.”

In summary, while I have several concerns and suggestions outlined above that the authors should address before publication in Nature Communications, I believe this work demonstrates significant breakthroughs in multiple areas—including quantum system gain engineering, topological lasing with bosonic atoms, and potential developments in non-Hermitian engineering. I look forward to reviewing the revised manuscript before making my final recommendation.

We greatly appreciate your positive feedback and insightful comments. We believe that this response has properly addressed all your questions.

Reviewer #2 (Remarks to the Author):

The authors realise Bose-Einstein condensation in a micro-wave dressing of the hyperfine levels of 87Rb atoms. They perform evaporative cooling in the dressed states after carefully preparing the initial thermal states. In this approach, the thermal atoms can be seen as a reservoir, which provides “gain” for a “laser” formed of condensate atoms. They demonstrate this experimental approach in two configurations with 3 sites and 5 sites respectively, where the second engineers a SSH model, one of the simplest topological models in 1d. There, condensation occurs in a dressed state, which is a topological edge state of the 5-site system. Condensation is observed and quantified by observing the momentum distribution in time-of-flight with a magnetic gradient ; fitting the bimodal distribution identifies the condensed atoms and the thermal atoms.

A novel aspect is the ability to engineer gain in a setup with ultracold atoms, where it is usually difficult to obtain, contrary to losses, which opens a new path to study non-Hermitian physics with both gain and losses. A theoretical analysis is provided to support the claims and the interpretation of the experimental data, which is based on solving the non-Hermitian Schrödinger equation for the condensate wavefunction, taking the thermal atoms as a gain medium (which empties as the condensate grows), in the presence of losses for the condensate.

We thank the accurate description and positive assessment of our paper.

Overall I appreciated the manuscript and the work, but I have several questions and remarks for the authors, that would need to be addressed before supporting publication :

1) I find that the denomination “topological atom laser” is a bit overselling and could be simplified to : Bose-Einstein condensation / Lasing in a topological edge state.

We appreciate your comment on the title and acknowledge your suggestion. We will update the title from “Gain engineering and topological atom laser in synthetic dimensions” to “Gain engineering and atom lasing in a topological edge state in synthetic dimensions”.

2) While similar experiments have been performed in photonic systems, such as in lattices of polariton micropillars realising a SSH model [ref 9 of the authors], this is the first observation in cold atoms.

In the polariton experiment, the system is pumped incoherently and above a critical threshold, lasing suddenly dominates in an edge mode. However, here, the authors explain lasing in the edge state on page 2, right column, by a careful preparation of the reservoir of thermal atoms. As the atoms are initially in the dark state due to the time-dependent microwaves, the population in the other states is indeed negligible, and thus we both only expect the dark state to be populated and the dark state to have a significant condensate fraction. This is obvious if there is no mechanism that would connect atoms in the dark state to a lower-energy state. In my opinion, this differs from the lasing demonstrated in other systems, where all modes are pumped and one wins above a critical threshold ; here, only the dark state is occupied by the thermal atoms which naturally forces Bose-Einstein condensation in this mode.

From the sentence “We see that in all three values of r , the BEC phase transition, namely, atom lasing is successfully observed, and the formed BEC has the largest population in site 5, with certain atom numbers in 3 and 1 with almost no occupation in 2 and 4, which is consistent with the wavefunction profile of the topological edge state of the SSH model.”, I do not see what else I should expect.

It would be interesting to explain what would happen if the thermal atoms are prepared in a state which is not dark for example and discuss the robustness of the scheme with the preparation of the thermal atoms in a particular state. A clearer explanation of the role of the dark state is necessary for me.

We fully agree with the reviewer that what happens if the thermal atoms are not prepared only in the dark state is an interesting problem. For the photonic systems, it is true that the pumped states do not precisely correspond to the edge states, but, during the lasing only the edge state experiences growth. This behavior is also confirmed in our numerical simulations of our atom experiment. For instance, in Fig. R1, we show the result of our simulation where the initial state population for the site 5 is 0.8, while those of the sites 1, 2, 3, and 4 are randomly distributed, which is not apparently the pure edge state, corresponding to the edge-state fidelity of approximately 0.9. All other simulation conditions remain essentially the same as those outlined in the main text. Although, at $t=0$, the edge state fidelity does not reach 1 but 0.9, upon introducing gain for $t>0$, the BEC emerges with the fidelity approaching 1, surpassing that of the thermal state.

Figure R1: Numerical simulation for the case of non-ideal population in the 5-site model. Note that the values of the gain, loss, and magnetic field fluctuations are the same as those shown in Figure 4, and the only difference is the initial state of the thermal atoms. When the total number of atoms is N_{total} , a fraction pN_{total} is allocated to site 5, whereas the remaining $(1-p)N_{\text{total}}$ atoms are randomly distributed among sites 1-4 at the initial stage. Time $t = 0$ ($t=50$ ms) marks the completion of state preparation (evaporative cooling). Φ_{sim} and Φ_{edge} denote the state obtained through time evolution in a numerical simulation and the expected edge state derived by diagonalizing equation (3) in the main text with $\delta = 0$, respectively. For a sufficiently large p (approximately $p > 0.8$), the BEC evolves into the same state as the edge state irrespective of the initial state.

It is important to note that this behavior is presumed to occur solely under the ideal conditions for numerical calculations. In actual experiments, significant magnetic field fluctuations were present, preventing the observation of situations predicted by numerical calculations. These discussions are elaborated upon in the Discussion section.

The newly added sentence reads

“What happens if the initial thermal atoms are not prepared only in the edge state, like the photonic systems, is an interesting problem. Our simulation where the initial state is not the pure edge state shows, although the fidelity does not reach 1 at $t=0$, upon introducing gain for $t>0$, the BEC emerges with the fidelity approaching 1, surpassing that of the thermal state. This is similar to the observation in the photonic systems. In our actual experiments, however, significant magnetic field fluctuations were present, preventing the observation of behaviors predicted by numerical calculations.”

3) The most natural description of such a system for me is the one of a spinor condensate. In the dressed picture, the eigenstates are decomposed over the hyperfine states which can interact at low temperature with s-wave collisions with interaction parameters that depend on the magnetic field and on the states. I did not see any discussion or comment on interactions parameters, which I would appreciate.

We thank the reviewer for the question. The scattering length of ^{87}Rb exhibits only a small dependence on hyperfine or magnetic sublevels [Widera et al, New J. Phys. 8, 152 (2006); Kaufman et al, Phys. Rev. A 80, 050701 (2009)]. As a result, the spin-exchange collisions are significantly suppressed. We have added this sentence in the main text.

The newly added sentence reads,

“The scattering length of ^{87}Rb exhibits only a small dependence on hyperfine or magnetic sublevels [Widera et al, New J. Phys. 8, 152 (2006); Kaufman et al, Phys. Rev. A 80, 050701 (2009)]. As a result, the spin-exchange collisions are significantly suppressed.”

4) Related to my 2nd comment, I would find instructive to see what are the populations in the different hyperfine states at the end of the partial STIRAP procedure and before evaporation, to be able to compare them with the observed populations after evaporation. For example, in figure 2b, I would like to understand if the population in the state 3 originates purely from imperfections in the STIRAP procedure. This could be compared to

the results of the theoretical model and figure 4d-e, which indicate that “that the edge state are adiabatically prepared”.

We thank the reviewer for this important question. Note that “Theory” in Figs. 2, 3, and 5(b) in the original manuscript corresponds to the numerical simulation in the idealized case of no detuning ($\delta = 0$), as well as the data in Fig. 4(e). While the numerical simulations shown in Fig. 4(a-d, f, g) were done in the presence of the detuning oscillating at 60 Hz, here we performed the numerical simulation in the presence of a static detuning due to the slow drift of a magnetic field, which breaks the chiral symmetry for the 5 site system and dark-state condition for the 3-site system. The case of the 3-site system in the presence of 0.25 mG static magnetic field is shown in Fig. R2 where the finite population in the state 3 at the end of the partial STIRAP procedure and before evaporation is clearly found. Similar finite populations in the even sites are found in the case of 5-site system in the presence of 0.5 mG, as shown in Fig. R3. Note that, in these conditions, dark state and edge state fidelities of the subsequent BEC do not surpass those for the thermal atoms at $t=0$, contrary to the case of no static detuning but with the finite populations in the site 3 (3-site system) and even sites (5-site system), indicating the importance of the chiral symmetry and dark-state condition in the behavior of atom lasing.

The newly added sentence reads,

“We also performed the numerical simulation in the presence of a static detuning due to the slow drift of a magnetic field, which breaks the chiral symmetry for the 5 site system and dark-state condition for the 3-site system. As expected, we found the finite populations in the state 3 and even sites in the case of 3- and 5-site systems, respectively, at the end of the partial STIRAP procedure and before evaporation.”

In the revised manuscript, we also explicitly note that “Theory” in Figs. 2, 3, and 5(b) in the original manuscript corresponds to the numerical simulation in the idealized case of no detuning ($\delta = 0$), as well as the data in Fig. 4(e), in the corresponding Figure Captions.

Figure R2: Numerical simulation in the presence of a static detuning due to a 0.25 mG static magnetic field in the 3-site model. The values of the gain $g=17.5$ and loss $\gamma=10$ are the same as those shown in Figure 2. The finite population in the state 3 is clearly found. The region $t < 0$ corresponds to the state preparation, that is, the Rabi coupling is gradually introduced to the thermal atoms. The region $t > 0$ corresponds to evaporatively cooling with gain for the growth of BEC. Note that the adiabaticity is slightly violated, causing an oscillatory behavior at $t > 0$.

Figure R3: Numerical simulation in the presence of a static detuning due to a 0.5 mG static magnetic field in the 5-site model. The values of the gain and loss are the same as those shown in Figure 4. The finite population in the even states of 2 and 4 is clearly found. The region $t < 0$ corresponds to the state preparation, that is, the Rabi coupling is gradually introduced to the thermal atoms. The region $t > 0$ corresponds to evaporatively cooling with gain for the growth of BEC. The adiabaticity is slightly violated, causing a slightly oscillatory behavior at $t > 0$.

5) In the 5-site SSH configuration, the condensate fraction are much smaller than in the 3-site configuration. Why is that ? The total condensate fraction that is reached is not written

in any case.

We thank the reviewer for the question. Compared to the 3-site system which mainly populates the lower hyperfine levels with negligible inelastic loss, the 5-site system mainly populates the upper hyperfine levels which suffer from larger inelastic collisions [63-66 in the revised manuscript] with more complex microwave couplings, making evaporative cooling more challenging. Information pertaining to the BEC fraction has been incorporated in Figures 2, 3 and 5.

The newly added sentence reads,

“Compared to the 3-site system which mainly populates the lower hyperfine levels with negligible inelastic loss, the 5-site system mainly populates the upper hyperfine levels which suffer from larger inelastic collisions [63-66] with more complex microwave couplings, making evaporative cooling more challenging.”

6) On page 5, equation 4 : the coefficient g is not explained. How can it be deduced from a microscopic analysis ?

We thank the reviewer for the question. As mentioned in the main text, the gain coefficient g is a phenomenological parameter. While we do not know at the moment how to microscopically derive this parameter, what we can say is that the gain is related with the microscopic process of bosonic stimulation through atom collisions and thus should be proportional to the number of thermal atoms, which is how we constructed the theoretical model.

The newly added sentence reads,

“Note that the gain is related with the microscopic process of bosonic stimulation through atom collisions and thus, the gain coefficient g , introduced as a phenomenological parameter, should be multiplied by the number of thermal atoms, which is how we constructed the theoretical model. “

7) On page 6, figure 4g, can the fidelity be accurately extracted at times $t < 30\text{ms}$, where the population in the BEC state is actually very close to zero ? It seems to me that the fidelity is above 0.95 as soon as the population is non negligible.

Related to my 4th comment, what is the expected fidelity for the thermal component, especially at $t=0$ at the end of the MW preparation? What is the sensitivity of the lasing in the dark state on the MW preparation based on the theoretical model?

We thank the reviewer for this important question. We here explain the details of behavior shown in Fig. 4 (g). As one can see in Fig. 4(a), (c), and (f), we set very small but finite “BEC” populations in the initial state as a “seed” for BEC. As the gain is introduced for $t>0$, the population of BEC in an edge state increases, approaching unity. We agree that the relative populations and “fidelity” for the BEC is only important when the BEC fraction is increased to a certain amount beyond the initial “seed” necessary in the numerical calculation.

The relative populations of the thermal atoms at $t=0$ depend on how adiabatic the state preparation, or the microwave sweep, is. The results of the numerical simulations showed that the sweep time of 10 ms is sufficiently adiabatic. As was also mentioned in the reply for the 4th comment, the numerical simulation in the presence of a static detuning due to the slow drift of a magnetic field of 0.5 mG, which breaks the dark-state condition for the 3-site system, shows that the finite population in the state 3 at $t=0$ is clearly found.

The newly added sentence reads,

“The relative populations and fidelity for the BEC shown in Fig. 4 (c), (f), and (g) are only important when the BEC fraction is increased to a certain amount beyond the initial “seed” necessary in the numerical calculation.”

8) On page 7, the part on “Confirming the coherence of the topological atom laser”, along with figure 5, is important in my opinion and I appreciated it. How is the fidelity 0.95(5) estimated? It is not obvious for me based on the data of figure 5 and the relative population in state 5 below 0.8.

We apologize for the unclear explanation. First, the definition of fidelity used here is the first power of the inner product and not the second power. In addition, there are two methods to perform the required calculations. Originally we estimated the value of fidelity using a series of absorption images. The first method is to calculate the fidelity of each image and then averages the values. The second method is to average the number of atoms at site i ($i=1\cdots 5$) and then calculate the fidelity. The first method gave a result of 0.95(5), and the second method gave 0.90(1). In the revised manuscript, we chose the second

method and used 0.90(1) for more appropriateness and clarity. We would like to note that the occupancy ratio of site 5 was 0.82(2), which is about the square of Fidelity of 0.90(1).

To avoid the possible confusion, we newly added the following sentence,
“...the quantity of $I=0.90(1)$ for the condensates, consistent with 0.82(2) in terms of population, shown in Fig. 5 c. This value of I close to 1 indicates ...”

Additional smaller comments :

1) In the figures 2 and 3, are the fourths rows commented in the main text ? I did not see their use. It seems to me that the sentence beginning by “In the third (fourth) row of Fig. 2,” on page 2, right column, does not exactly matches the content of the figure.

We apologize for the unclear wordings and missing connection with the figures. We have revised the text in the manner in which the figures were properly referenced.

See revised Fig. & Fig. caption

2) On page 3, right column : typo “As we see in in Methods”

We thank the reviewer for your pointing out this mistake. We have made necessary corrections.

3) On page 3, equation 3 : the indices (1,2) are not explicited for the Rabi frequencies in microwave coupling

We are sorry that we have not explained these in detail. Thank you for pointing these out to us. We have added an explanation in Figure 1.

The newly added sentence reads,

“Rabi frequencies for the microwave coupling are indicated by ... and $\Omega_i^{(j)}$ ($i=S,W$, $j=1,2$) for 5-site lattice”

4) On page 4, right column, the reasons for the “not a simple ascending order” could be explicitly explained in a short way, based on the difference in sign of the magnetic moments of states of same projection m_F between the $F=1$ and $F=2$ states.

We thank the reviewer for the comment. We have added a sentence to explain why the atom positions in the images after time-of-flight are not in a simple ascending order.

The revised sentence now reads,

“The order of sites 1, 4, 3, 2, 5 in the absorption image is not a simple ascending order because of the difference in the sign of the magnetic moments of the states between the $F=1$ and $F=2$ states.”

5) The sentence with the definition of H_{gain} on page 5 left column does not have a verb.

Thank you for your pointing out this mistake. We have made necessary corrections.

The revised sentence now reads,

$\hat{H}_{\text{gain}} = \text{ignT} |T\rangle \langle T|$ represents the gain term with the coefficient g being the effective strength of the gain per unit thermal component.

Reviewer #3 (Remarks to the Author):

Summary

The authors report the experimental observation of BEC formation in excited eigenstates of a synthetic lattice of hyperfine states of ^{87}Rb atoms. Their gain engineering scheme is original and well described, and supported by a theoretical model. They further employ their scheme to engineer a 5-site synthetic SSH chain, in which they promote lasing from topological edge modes. The technique presented in the article can catalyze the exploration of driven-dissipative phenomena in ultracold atomic gases.

General Remarks

The article is well detailed, both in terms of the experimental protocol and the theoretical modeling and simulations. I think that, overall, it warrants publication in Nature Communications. I do have, however, several remarks that I would like the authors to address.

We greatly appreciate the positive assessment of the reviewer.

1. The information in the article is quite scattered throughout the pages, and early pages and statements become clear only later on. The reader, therefore, has to hold on to several bits of information before they are eventually clarified, which in my opinion leads to a confusing reading experience. There are also several repetitions, for example the abstract and the first lines of the introduction, or in the “Discussions” section.

We are sorry for the improper structure of our manuscript. We have revised the manuscript in a manner that the provided information should not be scattered. In addition, we removed the repetitions in the Abstract, Introduction, and Discussion sections as possible as we can. Because of the numerous sections that have been updated, we will detail them in “Summary of Changes”.

2. What is the rationale behind making an atom laser topological, especially an SSH (or 1D, in general) one? What would be the advantages (and disadvantages) over a conventional atom laser?

We thank the reviewer for the question. The purpose of this paper is proof of principle, but the topological atom laser should share the merits the topological photonic lasers have, such as the robust single-mode lasing (condensation) in the presence of disorder. However, we want to stress that the main technological implication of our paper is the ability to introduce gain in ultracold gases, which allows one to explore a variety of non-Hermitian Hamiltonians in ultracold settings.

The newly added sentence reads,

“While the main technological implication of our work is the ability to introduce gain in ultracold gases, which allows one to explore a variety of non-Hermitian Hamiltonians in ultracold settings, the topological atom laser should share the merits the topological photonic lasers have, such as the robust single-mode lasing (condensation) in the presence of disorder.”

3. Also in connection with my previous question, the authors show BEC formation, but what about lasing characteristics? What’s the threshold behavior? What are the spectral

properties? What's the second order coherence?

We thank the reviewer for this interesting question. What we mean by the atom laser is, following existing literatures, the Bose-Einstein condensation. It is difficult to find exact correspondence of lasing characteristics discussed in optical lasers, although we believe that they are promising subjects for future research. Second order coherence perhaps corresponds to two-point correlation function of Bose-Einstein condensation, but we believe there is nothing special about topological atom laser in terms of these characteristics compared to the conventional Bose-Einstein condensation.

The newly added sentence reads,

“Note that we believe there is nothing special about topological atom laser in terms of characteristics such as threshold behavior, spectral properties, and the second order coherence compared to the conventional Bose-Einstein condensation, although they are subjects for future research.”

4. This is more of a curiosity, not a request for review; would it be possible to directly measure the Zak phase in this particular synthetic lattice?

We thank the reviewer for this interesting question. Zak phase is a property of bulk, and in our experiment there are only 2.5 unit cells, which makes it difficult to probe bulk properties. In order to explore bulk topological physics we need to use a longer lattice, which is an interesting future work.

Specific Remarks

1. Just before eq. 1, the authors write “ λ -type”, although later they use “ Λ -type” and “W-type”. Please use a consistent notation.

Thank you for your pointing out this inconsistency. In the revised manuscript, we use “ Λ -type”, and “W-type” for the 3-site and 5-site systems, respectively.

2. At the beginning of the right column of page 2, they authors write “ τ / τ_{full} leads to the

equal superposition of site 4 and 2". While this seems to be clear from their previous expression for the dark state (at $\delta=0$) and from the experimental results of Fig. 2b, third row, the "Theory" panel on the same row seems to predict a non-symmetric distribution. Why is that? Does it also account for a non-zero δ ? And if so, why the population on site 3 is still zero?

We thank the Reviewer for the questions. We apologize for these potentially misleading expressions. As shown in Table I, Ω_1 and Ω_2 are not equal when $\tau / \tau_{\text{full}} = 0.5$. In the 3-site system, we believe that labeling with τ is not appropriate. In the revised manuscript, we have labeled using the Ω_1 to Ω_2 ratio. The time over which the values change adiabatically is newly presented in Table I. In this revised manuscript, we show three cases where Ω_1/Ω_2 is much less than 1, approximately equal to 1, or much greater than 1, and we believe that it is a more reader-friendly way of presenting the experimental parameters.

We again note, as in the reply of 4th comment of reviewer#2, that the "Theory" in Figs. 2, 3, and 5(b) in the original manuscript corresponds to the numerical simulation in the idealized case of no detuning ($\delta=0$), as well as the data in Fig. 4(e).

3. Again in Fig. 2, the legend text in the bottom row is way too small, and the remaining text could also be made a little bigger to improve legibility. If necessary, the authors can make more space by collapsing repeated x- and y-labels. This also applies to Fig. 3 and 1.

Thank you for your pointing out these issues. We have made necessary corrections.

4. Continuing the discussion of Fig. 2, why the a) and c) columns are show for $\tau / \tau_{\text{full}} = 0.3$ and 0.8? What happens at $\tau / \tau_{\text{full}} = 0$ and 1? Aren't the "Theory" distributions predicted for those values?

We thank the Reviewer for the questions. We did not have experimental data corresponding to $\tau / \tau_{\text{full}}=0$ or $\tau / \tau_{\text{full}}=1$ in the original manuscript. $\tau / \tau_{\text{full}}=0$ and $\tau / \tau_{\text{full}}=1$ are almost the same as $\tau / \tau_{\text{full}}=0.3$ and $\tau / \tau_{\text{full}}=0.8$, respectively, and the differences are only trivial. As mentioned in our answer to Specific Remark 2), we believe that labeling with $\tau / \tau_{\text{full}}$ is

not appropriate. Therefore, in the revised manuscript, we used Ω_1/Ω_2 for labeling. We believe this is a less misleading way to express it.

5. Let's go back to Fig. 1. Why the site ordering 1-4-3-2-5 in column c? In the same column, are Ω_S and Ω_W the averages defined later in the text? Shouldn't the 5-site frequencies in the left panel be marked as $\Omega_S^{(1,2)}$ and $\Omega_W^{(1,2)}$?

We thank the reviewer for the questions. The first question is the same as the question (4) by Reviewer 1. We have added a sentence to explain why the atom positions in the images after time-of-flight are not in a simple ascending order.

The revised sentence now reads,

“The order of sites 1, 4, 3, 2, 5 in the absorption image is not a simple ascending order because of the difference in the sign of the magnetic moments of the states between the $F = 1$ and $F = 2$ states.”

As for the second question, we revised Fig. 1 so that the 5-site frequencies are marked as $\Omega_S^{(1,2)}$ and $\Omega_W^{(1,2)}$, according to the suggestion. The averages of $\Omega_S^{(1,2)}$ and $\Omega_W^{(1,2)}$, marked as $\langle \Omega_S \rangle$ and $\langle \Omega_W \rangle$ are only defined later in the text (“Experimental results”).

6. Why is there a large mismatch wrt the theoretical prediction in the rightmost panel of the third row in Fig. 3?

We thank the reviewer for the question. The rightmost panel of the third row in Fig. 3 in the original manuscript corresponds to the case of the largest microwave power to construct the edge state, beyond the regime of $\Omega_S^{(1,2)} > \Omega_W^{(1,2)}$, and thus the largest change of populations, possibly suffering from imperfections such as a magnetic field fluctuation.

The newly added sentence now reads,

“One can see a relatively large mismatch between the theoretical prediction and the data in the case of the largest microwave power, beyond the regime of $\Omega_S^{(1,2)} > \Omega_W^{(1,2)}$. Since this

case corresponds to the largest change of populations, and thus possibly suffers relatively much severely from imperfections such as magnetic field fluctuation.”

7. In the “Discussions” section, it’s stated again that a small occupation in sites 2 and 4 is to be expected due to the non-zero magnetic field fluctuations δ that break the chiral symmetry. While this seems to be incorporated in the theory, the “Theory” panels throughout the text have identically zero occupation in the even sites. Why is that?

We thank the reviewer for this important question. Please see the answer for the Reviewer #2’s question (4). “Theory” in Figs. 2, 3, and 5(b) in the original manuscript corresponds to the numerical simulation in the idealized case of no detuning ($\delta = 0$), as well as the data in Fig. 4(e). The numerical simulations shown in Fig. 4(a-d, f, g) were done in the presence of the small detuning oscillating at 60 Hz. Here we performed the numerical simulation in the presence of a static detuning due to the slow drift of a magnetic field, which breaks the chiral symmetry for the 5 site system. The case of 0.5 mG static magnetic field is shown in Fig. R3 where the finite population in the even states at the end of the partial STIRAP procedure and before evaporation is clearly found.

As is already noted in the reply for Reviewer #2’s Question (4), the corresponding newly added sentence reads,

“We also performed the numerical simulation in the presence of a static detuning due to the slow drift of a magnetic field, which breaks the chiral symmetry for the 5 site system and dark-state condition for the 3-site system. As expected, we found the finite populations in the state 3 and even sites in the case of 3- and 5-site systems, respectively, at the end of the partial STIRAP procedure and before evaporation.”

We also explicitly note that “Theory” in Figs. 2, 3, and 5(b) in the original manuscript corresponds to the numerical simulation in the idealized case of no detuning ($\delta = 0$), as well as the data in Fig. 4(e), in the corresponding Figure Captions.

8. In relation to my previous answer, why do the authors choose the loss coefficient they way they do just before equation 9? There doesn’t seem to be any justification for their choice. This leads to the loss Hamiltonian in equation 9, which is 0 on sites 2 and 4. Could this be the source of the identically zero population on sites 2 and 4 in the “Theory” panels

throughout the text? Wouldn't this choice make the non-zero condensate density solution in sites 2 and 4 locally (dynamically) unstable, and thus forbidden in the numerical solutions?

We thank the reviewer for this important question. We do not know all the loss coefficients. The loss Hamiltonian in Eq. (9) based on the reported measurement results is adopted in this work.

As was explained in the reply for the previous question as well as that for the reviewer #2's question (4), "Theory" in Figs. 2, 3, and 5(b) in the original manuscript corresponds to the numerical simulation in the idealized case of no detuning ($\delta = 0$), as well as the data in Fig. 4(e), which is the source of the identically zero population on sites 2 and 4 in the "Theory" panels of Fig. 3, as well as the site 3 of Fig. 2.

As for the numerical simulation including the loss and gain terms, as shown in Fig. 4(a-d, f, g), only when we introduce a static detuning, we find a finite population in the site 3 (3-site system) and even-sites (5-site system), which illustrates the importance of the chiral symmetry and dark-state condition to suppress the unwanted populations. In this respect, a specific form of loss Hamiltonian does not play an important role in the current work.

The newly added sentence reads,

"The choice of this specific form of loss Hamiltonian of Eq. (9) is based on the reported measurement results, and does not have a direct relation to the appearance or suppression of the populations in the even sites where the loss is set to zero."

9. The legends in the last row of Fig 5 are again too small.

We thank the reviewer for this pointing out. We deleted the legend from the image and explained in the figure caption because of limited available space.

Summary of Changes:

1. We changed the title according to the advice of the Reviewer #2. The new title is “Gain engineering and atom lasing in a topological edge state in synthetic dimensions”.
2. We modified the Abstract section in order to remove repetition with respect to the “INTRODUCTION” and “DISCUSSION” sections.
3. We added the sentence “The scattering length of ^{87}Rb exhibits only a small dependence on hyperfine or magnetic sublevels [Widera et al, New J. Phys. 8, 152 (2006); Kaufman et al, Phys. Rev. A 80, 050701 (2009)]. As a result, the spin-exchange collisions are significantly suppressed.” in the “EXPERIMENTAL RESULTS” section.
4. We added the sentence “Note that the gain is related with the microscopic process of bosonic stimulation through atom collisions and thus, the gain coefficient g , introduced as a phenomenological parameter, should be multiplied by the number of thermal atoms, which is how we constructed the theoretical model.” in the “THEORETICAL MODEL” section.
5. We added the sentence “The relative populations and fidelity for the BEC shown in Fig. 4 (c), (f), and (g) are only important when the BEC fraction is increased to a certain amount beyond the initial “seed” necessary in the numerical calculation.” in the “THEORETICAL MODEL” section.
6. We added the sentence ““...the quantity of $I=0.90(1)$ for the condensates, consistent with $0.82(2)$ in terms of population, shown in Fig. 5 c. This value of I close to 1 indicates ...” in the “CONFIRMING THE COHERENCE OF THE TOPOLOGICAL ATOM LASER” section.
7. We added the sentence “We also performed the numerical simulation in the presence of a static detuning due to the slow drift of a magnetic field, which breaks the chiral symmetry for the 5 site system and dark-state condition for the 3-site system. As expected, we found the finite populations in the state 3 and even sites in the case of 3- and 5-site systems, respectively, at the end of the partial STIRAP procedure and before evaporation.” in the “DISCUSSION” section.

8. We added the sentence “What happens if the initial thermal atoms are not prepared only in the edge state, like the photonic systems, is an interesting problem. Our simulation where the initial state is not the pure edge state shows, although the fidelity does not reach 1 at $t=0$, upon introducing gain for $t>0$, the BEC emerges with the fidelity approaching 1, surpassing that of the thermal state. This is similar to the observation in the photonic systems. In our actual experiments, however, significant magnetic field fluctuations were present, preventing the observation of behaviors predicted by numerical calculations.” in the “DISCUSSION” section.

9. We added the sentence “Note that non-Hermitian dynamics can be realized through the demonstrated gain engineering and atom loss. For example, if we prepare thermal atoms populated at sites 1, 3, and 5, and introduce dissipation at sites 2 and 4, we should be able to realize a PT-symmetric non-Hermitian system. Such models should feature PT-symmetric/broken phase transition, which should be observable within current experimental techniques.” in the “DISCUSSION” section.

10. We added the sentence “Here, the subsystem which can be described in terms of the non-Hermitian Hamiltonian is the Bose-Einstein condensate part of the atoms. The “external” bath which acts to provide the gain is the thermal (non-condensed) atoms. By adding spin-dependent gain and loss in the spin-orbit coupled ultracold gases [Ren et al., Nat. Phys. 18, 385-389 (2022); Guo et al., Phys. Rev. A 106, L061302 (2022)], for example, one should also have the non-Hermitian skin effect, characteristic of a non-Hermitian system.” in the “DISCUSSION” section.

11. We added the sentence “Note that we believe there is nothing special about topological atom laser in terms of characteristics such as threshold behavior, spectral properties, and the second order coherence compared to the conventional Bose-Einstein condensation, although they are subjects for future research.” in the “CONCLUSION AND PROSPECTS” section.

12. We added the sentence “Since there is no Bose-Einstein condensation for fermions, our work cannot be directly applied to ultracold fermions. Lasing is a phenomenon specific to bosons. Gain engineering should, however, work for fermionic condensates like Bardeen-Cooper-Schrieffer or molecular BEC states.” in the “CONCLUSION AND PROSPECTS” section.

13. We added the sentence “While the main technological implication of our work is the ability to introduce gain in ultracold gases, which allows one to explore a variety of non-Hermitian Hamiltonians in ultracold settings, the topological atom laser should share the merits the topological photonic lasers have, such as the robust single-mode lasing (condensation) in the presence of disorder.” in the “CONCLUSION AND PROSPECTS” section.

14. We added the sentence “The choice of this specific form of loss Hamiltonian of Eq. (9) is based on the reported measurement results, and does not have a direct relation to the appearance or suppression of the populations in the even sites where the loss is set to zero.” in the “METHOD” section.

15. We added the sentence “Rabi frequencies for the microwave coupling are indicated by \dots and $\Omega_i^{(i)}$ ($i=S,W, j=1,2$) for 5-site lattice” in the caption of Figure 1.

16. We added the sentence “The order of sites 1, 4, 3, 2, 5 in the absorption image is not a simple ascending order because of the difference in the sign of the magnetic moments of the states between the $F = 1$ and $F = 2$ states.” in the caption of Figure 1.

17. We added the sentence “Compared to the 3-site system which mainly populates the lower hyperfine levels with negligible inelastic loss, the 5-site system mainly populates the upper hyperfine levels which suffer from larger inelastic collisions [63-66] with more complex microwave couplings, making evaporative cooling more challenging.” in the caption of Figure 3.

18. We added the sentence “One can see a relatively large mismatch between the theoretical prediction and the data in the case of the largest microwave power, beyond the regime of $\Omega_S^{(1,2)} > \Omega_W^{(1,2)}$. Since this case corresponds to the largest change of populations, and thus possibly suffers relatively much severely from imperfections such as magnetic field fluctuation.” in the caption of Figure 3.

19. We changed to use Ω_1/Ω_2 to denote the experimental conditions for the 3-site system, rather than τ .

20. We changed to use $\langle \Omega_S \rangle / \langle \Omega_W \rangle$ to denote the experimental conditions for the 5-site system, rather than r .

21. We revised the manuscript to improve readability, such as increasing the font size and avoiding the use of expressions such as "Top" and "Bottom." for Figures 1, 2, 3 and 5.

22. We added the information of BEC fractions for Figures 2, 3, and 5.

In addition, we corrected as many errors and typos as possible in the manuscript. For details, please refer to the attached file, starting from the next page. Items highlighted in red indicate deletions and those in blue indicate additions.